# Deciphering signaling mechanisms and developmental dynamics in extraembryonic mesoderm specification from hESCs

Baohua Niu[1,2,6], Da Wang [1,2,6], Yingjie Hu[1,2,6], Yundi Wang[1,2,6], Gaohui Shi [1,2,6], Zhongying Chen[3], Lifeng Xiang[4], Chi Zhang[1,2], Xuesong Wei[1,2], Ruize Kong[4], Hongzhi Cai[1,2], Weizhi Ji [1,2,5] ✉, Yu Yin [1,2,5] ✉, Tianqing Li[1,2,5] ✉ & Zongyong Ai [1,2,5] ✉

Extraembryonic mesoderm (ExM) is crucial for human development, yet its specification is poorly understood. Human embryonic stem cell (hESC)-based models, including embryoids and differentiated derivatives, are emerging as promising tools for studying ExM development. Despite this, the signaling mechanisms and developmental dynamics that underlie ExM specification from hESCs remain challenging to study. Here, we report that the modulation of BMP, WNT, and Nodal signaling pathways can rapidly (4-5 days) and efficiently (~90%) induce the differentiation of both naive and primed hESCs into ExM-like cells (ExMs). We reveal that ExM specification from hESCs predominantly proceeds through intermediates exhibiting a primitive streak (PS)-like gene expression pattern and delineate the regulatory roles of WNT and Nodal signaling in this process. Furthermore, we find that the initial pluripotent state governs hESC-based ExM specification by influencing signal response, cellular composition, developmental progression, and transcriptional characteristics of the resulting ExMs. Our study provides promising models for dissecting human ExM development and sheds light on the signaling principles, developmental dynamics, and influences of pluripotency states underlying ExM specification from hESCs.

The extraembryonic mesoderm (ExM) is an integral part of the amnion, yolk sac, and chorion. It is involved in the formation of blood islands and the umbilical cord; it also produces abundant extracellular matrix to support embryo development and may serve a structural function in the re-expansion of the conceptus after penetration into the maternal endometrium[1]. A recent study reported that the ExM derived placental mesenchymal cells, Hofbauer cells, erythrocytes, and endothelial cells[2]. Therefore, the ExM plays a crucial role in embryonic development. In rodents, the ExM is derived from gastrulating cells at Carnegie stage (CS) 6. By contrast, the human ExM is specified prior to gastrulation and initially emerges in peri-implantation embryos at CS5[1]. Due to ethical concerns and technical barriers in studying human peri-implantation development, as well as the lack of models, the molecular characteristics, developmental dynamics, and specification principles of human ExM have long been unclear.

[1]State Key Laboratory of Primate Biomedical Research; Institute of Primate Translational Medicine, Kunming University of Science and Technology, Kunming, Yunnan, China. [2]Yunnan Key Laboratory of Primate Biomedical Research, Kunming, Yunnan, China. [3]Ultrasound Department, The Third People's Hospital of Yunnan Province, Kunming, Yunnan, China. [4]Department of Reproductive Medicine, The First People's Hospital of Yunnan Province, Kunming, Yunnan, China. [5]Yunnan Provincial Academy of Science and Technology, Kunming, Yunnan, China. [6]These authors contributed equally: Baohua Niu, Da Wang, Yingjie Hu, Yundi Wang, Gaohui Shi. ✉e-mail: wji@lpbr.cn; yiny@lpbr.cn; litq@lpbr.cn; aizy@lpbr.cn

Recently, human ExM was characterized by marker staining[3], single-cell RNA sequencing (scRNA-seq)[3,4], and spatial transcriptomics[5,6] in 3D-cultured peri-implantation embryos and in vivo gastrulating embryos. Further, ExM specification was also observed and identified in the newly developed human post-implantation embryo models[3,7–11]. In addition, through the isolation of CDH1− mesenchymal by-product cells in naive human embryonic stem cell (hESC)-derived human trophoblast stem cell (hTSC) cultures by day 30 of conversion, Pham et al. successfully obtained purified ExM-like cells (ExMs) that could be expanded in hTSC medium (hTS-M, ASECRiAV) for over 14 passages (70 days)[12]. This study made it clear that naive hESCs can differentiate into ExMs, provided a valuable model to study human ExM specification, and also implied that the pre-implantation epiblast (Pre-EPI, the in vivo counterpart of naive hESCs) may be one of the developmental origins of pre-gastrulation ExM. By activating WNT signaling, Wang et al. induced the differentiation of primed hESCs into ExMs, which can further develop into mesenchymal stem cells[13]. Although a series of recent studies have made significant progress, our current understanding of human ExM specification is still very limited, and the following scientific questions have not been explored: (1) can hESCs respond to exogenous signals and differentiate efficiently into ExMs within a few days? because human ExM arises as early as embryonic day (E) 11-12, when its possible origins, the epiblast and hypoblast, have just segregated for around 5 days[1,3,14]; (2) what are the signaling mechanisms and specification dynamics underlying ExM specification from hESCs? and (3) for ExM specification, do epiblast cells at distinct embryonic stages exhibit spatiotemporal alterations in their signal response, developmental progression, and the cellular features of the resulting ExMs? Given the inaccessibility of human embryos, the aforementioned third scientific question cannot be addressed through the study of human embryos. Naive and primed hESCs, corresponding to Pre-EPI and post-implantation late epiblast (PostL-EPI) respectively[15,16], serve as the optimal in vitro models for simulating human epiblast development at different stages[17–19], and thus are well-suited to decode the third scientific question.

To address these scientific questions, we herein establish efficient culture systems that induce the differentiation of both naive and primed hESCs into expandable ExMs within 4-5 days. Based on these systems, we reveal the signaling principles, specification dynamics, and developmental roadmap for the differentiation of both naive and primed hESCs into ExMs, and find that ExM specification undergoes a primitive streak-like intermediate (PSLI). Furthermore, we dissect the shared and distinct aspects between naive and primed hESCs in their specification to ExM. Our work provides promising models for studying human ExM development and advances the understanding of ExM specification from hESCs.

## Results

### Rapid and efficient differentiation of naive hESCs into expandable ExMs

Expandable human ExMs have previously been successfully isolated by enriching a small amount of E-Cadherin-negative mesenchymal cells in naive hESC-derived hTSC cultures, but this conversion takes about a month[12]. We sought to establish an improved system to induce efficient differentiation of naive hESCs into expandable ExMs in a shorter period. AIC-N hESCs, established under normoxia, share transcriptional and epigenetic features with naive hESCs derived under 5% O$_2$ in t2iLGö, 5i/LAF or HENSM media[3]. We previously discovered that when inoculated onto feeders in the modified N2B27 medium containing FGF4 and heparin (termed FH-N2B27), continuous treatment with CHIR99021 (CHIR, an inhibitor of GSK3) and BMP4 induced AIC-N hESCs to predominantly differentiate into ExMs[3]. To generate expandable human ExMs, we inoculated dissociated AIC-N hESCs onto Matrigel-coated dishes in FH-N2B27 medium supplemented with CHIR and BMP4 (CB). After 4 days, these CB-treated AIC-N hESCs (CBNs)

rapidly lost their colony morphology and were converted into mesenchymal cells (Fig. 1a). Immunofluorescence (IF) staining showed that most cells were positive for ExM markers GATA6, SNAIL, VIM, KDR, and FLT1 in D4 CBNs, with few OCT4+T+, SOX17+, CK7+, HAVCR1+, TFAP2A+, TFAP2C+, E-Cadherin+, and FOXA1+ cells (Fig. 1b and Supplementary Fig. 1a). Flow cytometry data further confirmed that more than 90% of D4 CBNs expressed GATA6 and SNAIL (Supplementary Fig. 1b). We next compared the transcriptional states of D4 CBNs and AIC-N hESCs using bulk RNA-seq. Principal-component analysis (PCA) and clustering showed mutually exclusive groups of D4 CBNs and AIC-N hESCs (Fig. 1c and Supplementary Fig. 1c), suggesting a distinct change in cell identity. By analyzing differentially expressed genes between D4 CBNs and AIC-N hESCs, we observed a depletion of pluripotency markers such as *TFCP2L1*, *KLF17*, *DNMT3L*, *POU5F1*, *NANOG*, and *SOX2* in D4 CBNs, accompanied by significant upregulation of ExM markers such as *HAND1*, *KDR*, *FLT1*, *RSPO2*, *PDGFRA*, *GATA4*, *GATA6*, *FOXF1*, *DKK1*, *HOXA11*, and *NID2*[3,12] (Fig. 1d and Supplementary Fig. 1d). Consistent with previous findings[3,12], D4 CBNs also expressed epithelial-to-mesenchymal transition (EMT) markers *CDH2*, *VIM*, *TWIST1*, *SNAI2*, and *ZEB2*, as well as extracellular matrix (ECM) genes *LAMB1* and other collagen-related genes (Fig. 1d and Supplementary Fig. 1d). We further performed Gene Ontology (GO) enrichment and Kyoto Encyclopedia of Genes and Genomes (KEGG) pathway analyses of genes differentially expressed between D4 CBNs and AIC-N hESCs. Enriched GO terms in D4 CBNs included ECM, adhesion, cell migration, EMT, WNT, and BMP signaling pathways. KEGG pathway analysis revealed that D4 CBNs were enriched with genes for adhesion, ECM, regulation of actin cytoskeleton, PI3K–Akt, MAPK, WNT, Hippo, and TGF-β (BMP branch) signaling pathways (Fig. 1e). These results suggest that D4 CBNs have similar characteristics compared to previously reported ExMs in 3D-cultured human embryos[3] or those derived from naive hESCs[12], including the expression pattern of marker genes, signaling pathway activity, cell migration, and extracellular matrix secretion. The D4 CBNs from three different AIC-N hESC lines (AIC-N1, AIC-N2, and AIC-N4)[3] exhibited similar transcriptional signatures (Fig. 1c, d, Supplementary Fig. 1c).

To further characterize cell-type compositions, we performed single-cell RNA-sequencing (scRNA-seq) on D4 CBNs using the 10X Genomics platform (Supplementary Fig. 1e), and identified four major cell populations by Uniform manifold approximation and projection (UMAP) analysis (Fig. 1f). Cluster annotation was performed based on the expression of lineage-specific markers: the dominant ExMs (ExM, 86.1%), the subordinate ExMs expressing specific endoderm markers (EnExM, 7.4%), and a small number of amnion-like cells (AM1, 5.4%; AM2, 1%) (Fig. 1f, g). Both ExM and EnExM expressed ExM markers *CDH2*, *GATA6*, *KDR*, *GATA4*, *VIM*, *FLT1*, and *COL3A1*; ExM expressed greater levels of *KDR*, *GATA4*, *VIM*, *FLT1*, and *COL3A1*, whereas EnExM showed higher expression of *CDH2*, accompanied by expression of endoderm markers *AFP* and *APOA1*, but negligible *FOXA1/2*, *SOX17*, and *TTR* (Fig. 1g and Supplementary Fig. 1f). Amnion markers *IGFBP3*, *GATA3*, *TFAP2A*, *KRT7*, *WNT6*, and *ISL1* were all upregulated in both AM1 and AM2, with GABRP specific to AM1 while TP63 and ITGA6 to AM2. Notably, trophoblast markers *HAVCR1*, *GCM1*, and *CGA* were absent in both AM1 and AM2 (Fig. 1g and Supplementary Fig. 1f). Clustering revealed the similar identities of EnExM and ExM (Fig. 1h), and RNA velocity map showed that EnExM was progressing towards ExM (Fig. 1i), suggesting a transitional state of EnExM. Further, we compared scRNA-seq dataset of D4 CBNs with other datasets from human CS7 gastrulating embryo[4], cultured human peri-implantation embryos[3], and naive hESC derivatives in hTS-M[12]. By integration analysis, we observed that the ExMs in D4 CBNs were highly similar to those from both in vivo and in vitro human post-implantation embryos, but overlapped only with some early ExMs and were significantly separated from the late ExMs in hTS-M (Fig. 1j, k). The early ExMs are mainly from D13 and D18 ASECRiAV cultures, and the late

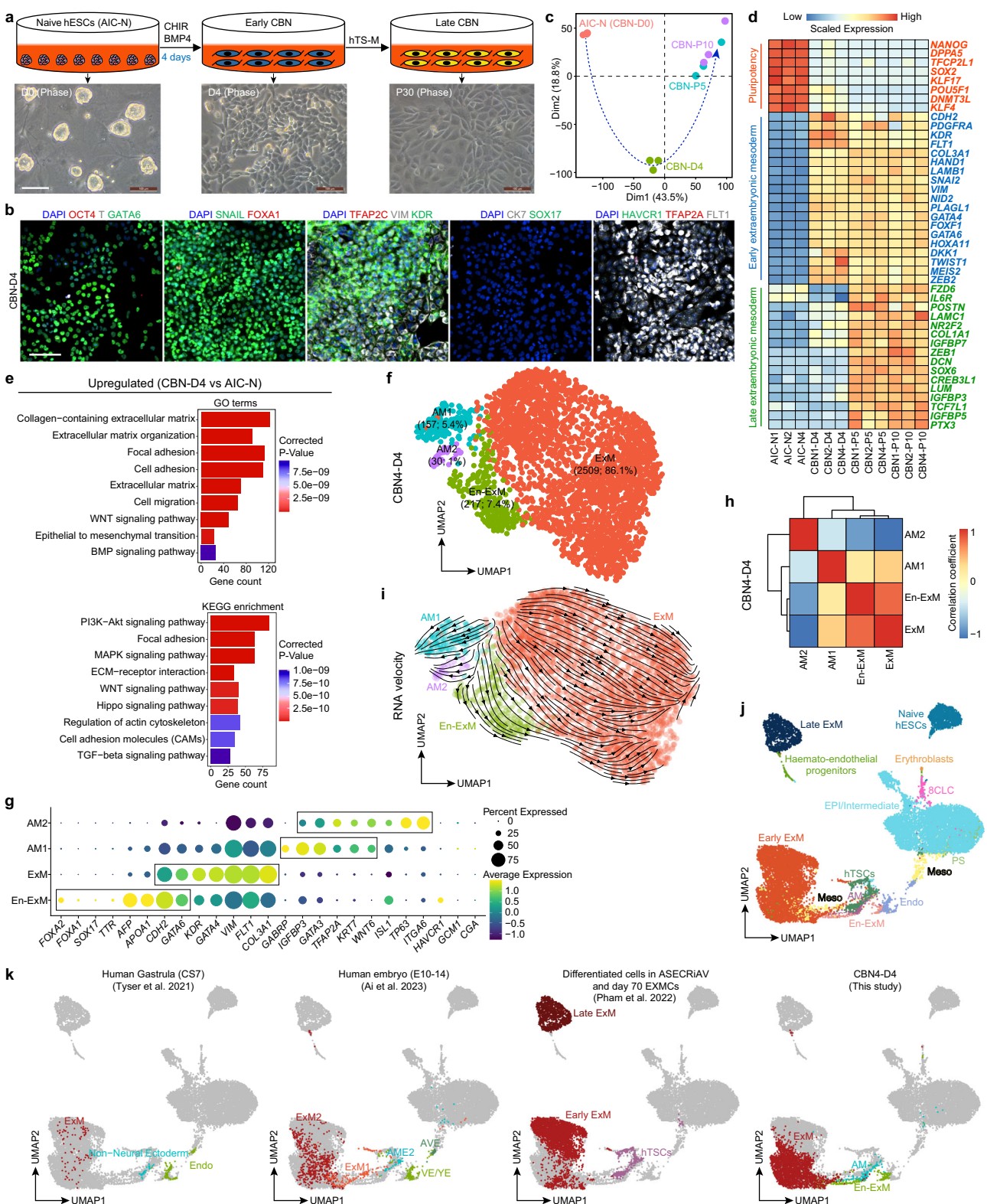

ExMs are purified from D30 ASECRiAV cultures and propagated for 70 days[12]. Therefore, most of the early and all late ExMs in hTS-M are at a more advanced developmental stage than those in D4 CBNs, potentially resulting in the separation of ExM clusters between the two datasets (Fig. 1j, k).

To determine if hTS-M supports self-renewal of our ExMs, we isolated ~90% of E-Cad⁻ cells from D4 CBNs by flow cytometry (Supplementary Fig. 1g) and seeded them to hTS-M (Fig. 1a and

Supplementary Fig. 1h). As expected, these E-Cad⁻ cells were able to self-renew and expand for more than 30 passages in hTS-M (Fig. 1a), with the expression of ExM markers such as GATA6, VIM, NR2F2, BST2, DCN, KDR, LUM, SNAIL, FLT1, and MEIS2 (Supplementary Fig. 1i), indicating the maintenance of ExM identity. PCA and clustering showed mutually exclusive groups of D4 CBNs and expandable ExMs, but expandable ExMs at different passages (P5 and P10) were clustered together, suggesting that the ExMs in D4 CBNs are in a transitional

**Fig. 1 | Naive hESCs rapidly produce expandable ExMs in response to WNT and BMP signaling. a** Schematic diagram and representative brightfield images demonstrating naive (AIC-N)[3] hESC derivatives induced by CHIR + BMP4 (CB) conditions, followed by extended culture in human trophoblast stem cellmedium (hTS-M). D, day; P, passage. **b** Immunofluorescence (IF) staining for various embryonic and extraembryonic lineage markers in CB-treated naive hESCs on day 4 (CBN-D4). **c** Principal-component analysis (PCA) of bulk RNA-seq data from differentiation time courses for CBNs, computed using the genes with FPKM ≥ 1 in at least one sample. **d** Heatmap of representative pluripotency and ExM marker genes in naive hESCs and CBNs. Values represent $\log_2$ (FPKM + 1) scaled by gene expression across samples. **e** Gene ontology (GO) and Kyoto Encyclopedia of Genes and Genomes (KEGG) analysis showing upregulated gene categories in CBN-D4 relative to naive (AIC-N) hESCs. Two-sided hypergeometric test (equivalent to Fisher's exact test), p-values were adjusted for multiple testing using the Benjamini-Hochberg method. **f** Uniform Manifold Approximation and Projection (UMAP) visualization of single-cell transcriptomics from CB-treated AIC-N4 hESCs on day 4 (CBN4-D4). UMAP plot is color-coded according to cell cluster identity annotations. ExM, extraembryonic mesoderm; AM, amnion; EnExM, ExM expressing specific endoderm markers. **g** Dot plots of candidate genes specific for cell subtypes. **h** Spearman correlation heatmap of different cell subpopulations, the colors indicate the Spearman correlation coefficient. **i** RNA velocity vectors projected onto the UMAP-based embeddings of the scRNA-seq dataset shown in (**f**). **j, k** UMAPs demonstrating the integration of CBN4-D4 with reference human embryonic cells and derivatives of naive hESCs in hTS-M (ASECRiAV). EPI epiblast, hTSCs human trophoblast stem cells, Endo endoderm, 8CLC 8-cell-like cell, Meso mesoderm, PS primitive streak, VE/YE visceral/yolk endoderm, AVE anterior visceral endoderm. All scRNA-seq data in this study are from 10× Genomics platform. Scale bars, 100 μm. Source data are provided as a Source Data file. See also Supplementary Fig. 1.

stage and progress to a stable state during their expansion in hTS-M (Fig. 1c and Supplementary Fig. 1c). Compared to D4 CBNs, expandable ExMs expressed higher levels of *FZD6*, *IL6R*, *LAMC1*, *COL1A1*, *IGFBP7*, *ZEB1*, *DCN*, *SOX6*, *CREB3L1*, *LUM*, *IGFBP3*, *TCF7L1*, and *IGFBP5*, as well as reported markers of late ExMs including *POSTN*, *NR2F2*, *PTX3*[12] (Fig. 1d), which may imply that expandable ExMs represent a more mature state. We further performed GO enrichment and KEGG pathway analyses of genes differentially expressed between D4 CBNs and expandable ExMs. Enriched GO terms in expandable ExMs included ECM, adhesion, and cell migration. KEGG pathway analysis showed that expandable ExMs were enriched with genes for focal adhesion, ECM−receptor interaction, regulation of actin cytoskeleton, calcium, PI3K−Akt, Jak−STAT, MAPK, cAMP, and VEGF signaling pathways (Supplementary Fig. 1j). These results suggest that hTS-M supports self-renewal and propagation of ExMs from D4 CBNs.

To assess the response of naive hESCs cultured under other medium to CB conditions, we transferred PXGL-hESCs[20] into CB conditions. Consistently, CB conditions promoted efficient differentiation of PXGL-hESCs into ExMs (Supplementary Fig. 1k−m). Together, CB conditions induce rapid and efficient specification of naive hESCs into expandable ExMs.

## WNT signaling is critical for ExM specification from naive hESCs

GO and KEGG analyses showed that WNT and BMP signaling pathways were enriched in D4 CBNs when compared with AIC-N hESCs (Fig. 1e). Consistently, many genes related to WNT and BMP signaling pathways were found upregulated in both D4 CBNs and expandable ExMs (Supplementary Fig. 2a). We next detected WNT and BMP signaling activity by IF staining for their key signal transducers: nuclear β-catenin for WNT and nuclear phosphorylated SMAD1/SMAD5/SMAD9 (pSMAD1/5/9) for BMP. We found that β-catenin was predominantly localized to the cell membrane and cytoplasm, with a complete absence of nuclear pSMAD1/5/9 in AIC-N hESCs (Supplementary Fig. 2b). In contrast, prominent nuclear pSMAD1/5/9 emerged in all D4 CBNs (Fig. 2a, b), indicating active BMP signaling. Notably, most of D4 CBNs, as putative GATA6+TFAP2A− ExMs, exhibited a clear nuclear accumulation of β-catenin (Fig. 2a, b), indicative of active WNT signaling. By contrast, in sporadic TFAP2A+ AM cells, although β-catenin displayed a higher level, it was predominantly located in the cell membrane and cytoplasm (Fig. 2a−c). Consistently, while *CTNNB1* had higher expression levels in AM1/AM2 than in ExM/EnExM, *AXIN2*, an indicator of canonical WNT activity[21], was only expressed in ExM/EnExM (Fig. 2d). The results suggest that β-catenin in AM cells likely functions predominantly as a component of adhesion junctions, interacting with cadherins such as E-cadherin to maintain cell−cell contacts[22], which is consistent with the upregulation of *CDH1* and the expression of E-cadherin in AM cells (Fig. 2d and Supplementary Fig. 1a). We further employed a TCF/LEF:H2B-GFP AIC-N hESC reporter line to generate D4 CBNs, and confirmed a low WNT activity in the majority of TFAP2A+ AM cells, where β-catenin predominantly localized to the cell membrane and cytoplasm (Fig. 2e, f).

Based on the results above, we reasoned that WNT inhibition may impair ExM differentiation while promoting AM specification in the context of BMP activation (Fig. 2g). Next, we tested the effect of WNT inhibition using IWP2 (an inhibitor of WNT protein secretion) in the presence of BMP4 (Fig. 2g). We observed that β-catenin mainly localized to the cell membrane and cytoplasm of IWP2 + BMP4-treated AIC-N hESCs (IBNs) on D4, and the vast majority of D4 IBNs exhibited negative or weak expression of TCF/LEF:H2B-GFP, indicating that WNT signaling is inhibited (Supplementary Fig. 2c). In contrast to D4 CBNs, almost all D4 IBNs expressed TFAP2C and E-Cad, with over 50% TFAP2A+HAVCR1− AM cells, but little to no KDR+VIM+ ExMs (Fig. 2h, Supplementary Fig. 2c−e), showing that the substitution of IWP2 for CHIR switched the fate of naive hESCs from ExM to AM. PCA analysis showed that D4 IBNs clustered closely with AIC-N hESCs, but separated from D4 CBNs (Fig. 2i), suggesting that D4 IBNs may contain some residual pluripotent cells, which was confirmed by gene expression and IF staining for pluripotency markers (Fig. 2j and Supplementary Fig. 2d). Notably, D4 IBNs upregulated AM-related genes while downregulating ExM-related genes compared to D4 CBNs (Fig. 2j). GO and KEGG enrichment analyses further showed that the genes downregulated in D4 IBNs were related to extracellular matrix, cell adhesion, EMT, and cell migration, as well as PI3K−Akt, MAPK, WNT, TGF-β (BMP branch), and Jak−STAT signaling pathways (Fig. 2k), revealing the loss of ExM identity. Collectively, these results demonstrate that, in the context of BMP activation, WNT signaling plays a switching role in the fate specification of AIC-N hESCs: WNT activation promotes ExM, while WNT inhibition favors AM (Fig. 2l).

## Specification dynamics of ExMs from naive hESCs

The specification dynamics and developmental roadmap of human ExM remain largely unclear. To reveal the differentiation trajectory of ExMs from AIC-N hESCs, we performed bulk RNA-seq analysis of CBNs and expandable ExMs across various differentiation time points. Clustering, correlation, and PCA analysis showed that AIC-N hESCs gradually transitioned from a pluripotent state to D3/D4 CBNs and ultimately to P5/P10 expandable ExMs (Fig. 3a, b). During the first two days of conversion, cells gradually downregulated the expression of pluripotency markers *KLF17*, *TFCP2L1*, *SOX2*, and *NANOG*, and upregulated the expression of primitive streak (PS) markers *MIXL1*, *CDX1*, *TBXT*, *SP5*, and *MESP1*, along with a modest upregulation of ExM markers such as *ZEB2*, *FLT1*, *VIM*, *HGF*, and *KDR* (Fig. 3c, d). From D3 onwards, the PS genes were rapidly downregulated, while the upregulation of ExM genes continued (Fig. 3c, d). IF staining further confirmed the expression dynamics of the pluripotency marker OCT4, the PS marker T (also known as Brachyury), and the ExM markers GATA6, FLT1 and KDR (Supplementary Fig. 3a, b). Throughout this

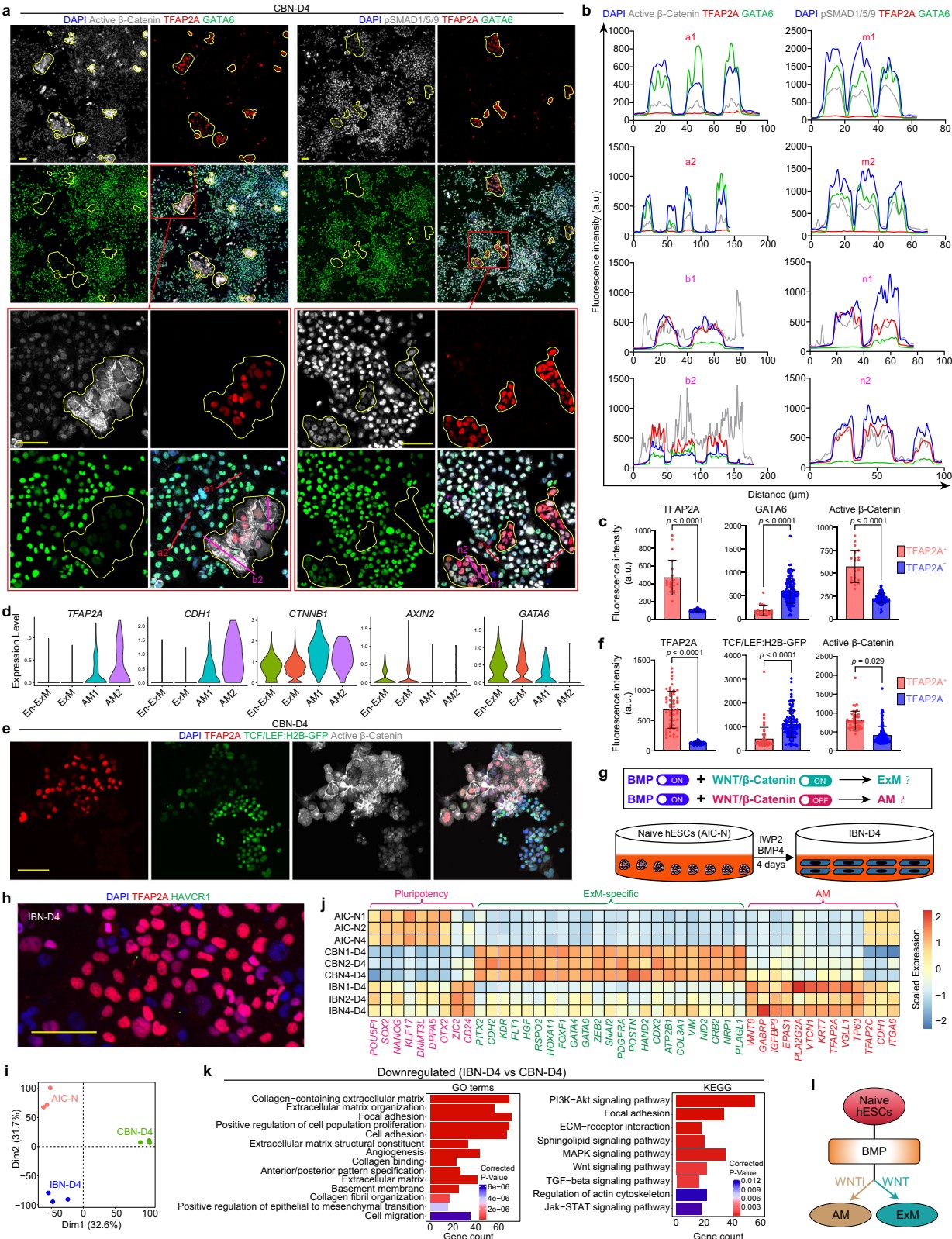

progression, there was no significant change in endodermal gene expression (Fig. 3c). We therefore speculated that ExMs in CBNs may predominantly originate from transient intermediates, exhibiting gene expression patterns similar to those of PS cells, which we referred to as primitive streak-like intermediates (PSLIs). These results may explain why ExM specification depends on WNT signaling (Fig. 2l), which is essential for driving PS-related gene expression[3,23–25].

To further explore the specification dynamics of ExMs from AIC-N hESCs, we performed 10X scRNA-seq on CBNs from D0 to D4 and produced an integrated UMAP embedding for CBNs at different time points (Fig. 3e, Supplementary Figs. 1e, 3c). Based on the expression patterns of lineage markers, we identified and characterized 8 clusters during the transition of CBNs, including Pre-EPI, transitional cell population (Tra), PSLI1/2, ExM1/2, primordial germ cells (PGC) and a

**Fig. 2 | ExM specification from naive hESCs depends on WNT signaling. a**, **b** IF staining (**a**) and fluorescence intensity profiles (**b**) for signal transducers and lineage markers in D4 CBNs. Red boxes mark the image areas for magnification. Yellow solid circles highlight TFAP2A⁺GATA6^weak/− AM-like cells. Red and pink lines indicate TFAP2A⁻ and TFAP2A⁺ planes used to plot intensity profiles of indicated markers, respectively. **c** Quantification of the mean fluorescence intensities of individual cells indicated in (**a**). Two-sided Student's *t*-test, *n* = 133 cells, three independent experiments. Data are presented as mean ± SD. **d** Violin plots of the indicated genes expressed in different types of cells by scRNA-seq data from D4 CBNs. **e** IF staining of AIC-N2 TCF/LEF:H2B-GFP hESC-derived D4 CBNs with indicated markers. **f** Quantification of the mean fluorescence intensities of individual cells indicated in (**e**). Two-sided Student's *t*-test, *n* = 166 cells, three experiments (technical replicates from AIC-N2 TCF/LEF:H2B-GFP hESC line). Data are presented as mean ± SD.

**g** Hypothetical model of the role of WNT/β-Catenin signaling (top) and schematic of IWP2 + BMP4 (IB) treatment in AIC-N hESCs (bottom). IBN-D4, IB-treated naive hESCs on day 4. **h** IF staining for AM and TrB markers in D4 CBNs. **i** PCA of bulk RNA-seq data from naive hESCs, CBN-D4, and IBN-D4, computed using the genes with FPKM ≥ 1 in at least one sample. **j** Heatmap of representative pluripotency, ExM, and AM marker genes in naive hESCs, CBN-D4, and IBN-D4. Values represent log₂ (FPKM + 1) scaled by gene expression across samples. **k** GO and KEGG analyses showing downregulated gene categories in IBN-D4 relative to CBN-D4. Two-sided hypergeometric test, *p*-values were adjusted using the Benjamini-Hochberg method. **l** Functional schematic of WNT and BMP signaling to specify ExM and AM lineages. Scale bars, 100 μm. Source data are provided as a Source Data file. See also Supplementary Fig. 2.

mixture of different types of cells (Mix) (Fig. 3e–h). Pre-EPI almost exclusively corresponded to AIC-N hESCs and expressed both naive and core pluripotency markers (Fig. 3e–h, Supplementary Fig. 3d). Tra, almost exclusively from D1 CBNs, was marked by a significant downregulation of anterior epiblast marker *SOX2* and a slight upregulation of PS marker *TBXT* (Fig. 3e–h, Supplementary Fig. 3d). The majority of PSLI1 cells were from D1 CBNs and a small percentage from D2 CBNs, PSLI1 was identified by loss of *SOX2*, downregulation of pluripotency marker *NANOG*, and significant upregulation of PS-specific genes *TBXT*, *MIXL1*, *SP5*, *HOXA1*, *CDX1*, *CDX2*, and *MESP1* (Fig. 3e–h, Supplementary Fig. 3d). PSLI2, mainly from D2 CBNs, lost the expression of *SOX2* and *NANOG*, downregulated the expression of *TBXT* and *MIXL1*, but further upregulated the expression of *SP5*, *HOXA1*, *CDX1*, *CDX2*, and *MESP1* (Fig. 3e–h, Supplementary Fig. 3d). Of note, in addition to the early ExM marker *FLT1*, PSLI2 also expressed *HAND1*, *GATA4*, *GATA6*, *CDH2*, and *VIM* (Fig. 3e–h, Supplementary Fig. 3d), all of which are markers of both PS and ExM[3]. ExM1 consisted of dominant D2 CBNs and secondary D3 CBNs, compared with PSLI2, ExM1 further upregulated *FLT1* expression but downregulated expression of PS-specific genes (Fig. 3e–h, Supplementary Fig. 3d). ExM2, the major cellular composition of D3 CBNs and D4 CBNs, further increased expression levels of *VIM* and *FLT1*, and upregulated expression of other ExM markers such as *KDR*, *COL3A1*, *COL6A1*, and *NID2* (Fig. 3e–h, Supplementary Fig. 3d). PGC expressing *TFAP2C*, *SOX17*, and *PRDM1* was mainly found in D2 CBNs, which was confirmed by IF staining of OCT4⁺SOX17⁺T⁺ and BLIMP1⁺NANOG⁺TFAP2C⁺ cells (Fig. 3e–h, Supplementary Fig. 3d, e). Mix cluster consisting of cells from D3 CBNs and D4 CBNs had a similar gene expression pattern to ExM2, but showed lower levels of ExM marker expression (Fig. 3e–h, Supplementary Fig. 3d). Within the Mix cluster, we further identified 3 subclusters with gene expression patterns similar to AM1, AM2, and EnExM described above (Fig. 1g, Supplementary Fig. 3f, g). Consistent with gene expression dynamics and time course of CBNs, the trajectory and pseudotime analysis further revealed a continuous developmental progression from Pre-EPI to Tra, then to PSLI, and finally to ExM (Fig. 3e, i, Supplementary Fig. 3d, h). We next performed GO and KEGG enrichment analysis to reveal signaling pathways involved in lineage transition of CBNs. Both Tra, PSLI1/2, and ExM1/2 clusters were enriched with genes associated with WNT and/or TGF-β (BMP branch) signaling pathways (Fig. 3h and Supplementary Fig. 3h), suggesting the key roles of WNT and BMP signaling in ExM specification. Moreover, GO and KEGG analyses of ExM1/2 clusters identified many enriched genes related to ECM, adhesion, basement membrane, cell migration, and PI3K−Akt signaling pathways (Fig. 3h and Supplementary Fig. 3h), indicating their ExM identity. These observations are consistent with bulk RNA-seq analysis results (Fig. 1e) and establish consistency with findings of ExM in the cultured human embryos[3].

Together, these data demonstrate that ExMs from AIC-N hESCs are largely specified through an intermediate, which recapitulates a gene expression program also involved in PS formation (Fig. 3j).

## PSLIs are specified prior to ExM formation in cultured human embryos

To further explore the developmental progression of human ExM, we integrated scRNA-seq data from cultured human embryos and CBNs across different time course and yielded highly congruent UMAP outputs (Fig. 4a). Correlation analysis and marker gene expression patterns exhibited prominent similarities between cultured human embryos and CBNs in the following cluster pairs: PostE-EPI versus Tra, PS1 versus PSLI1, PS2 versus PSLI2, ExM1 (embryo) versus ExM1 (CBN), and ExM2 (embryo) versus ExM2 (CBN) (Fig. 4a–c). Pseudotime analysis of cultured human embryos showed a developmental trajectory where pluripotency genes were gradually downregulated, PS genes first increased and then decreased, and ExM genes continued to be upregulated, consistent with the results observed in CBNs (Fig. 3i and Supplementary Fig. 4a). We further split the integrated UMAP data according to different developmental time points and found that the cultured human embryos are similar to the CBNs in terms of specification dynamics and developmental route of ExMs (Fig. 4d–f, Supplementary Fig. 4b, c). Both in cultured human embryos and CBNs, the PS genes *TBXT* and *SP5* were upregulated first, followed by the upregulation of PS gene *MESP1*, and finally the ExM genes *FLT1*, *HGF* and *COL3A1* (Fig. 4d–f, Supplementary Fig. 4b, c). Notably, the PS markers *TBXT* and *SP5* were expressed in some PostE-EPI cells as early as E10, while the expression of early ExM genes *FLT1*, *HGF*, and *COL3A1* remained undetected at this stage (Fig. 4d–f). These results indicate that a subset of cells expressing PS marker genes, potentially acting as PSLIs, are specified in cultured human embryos prior to ExM formation.

In cultured human embryos, we previously detected abundant ExMs by IF staining at E13[3]. To examine whether the emergence of potential PSLIs precedes ExM specification, we immunostained cultured human embryos at E11. Near OCT4⁺ epiblast compartment in two out of four embryos, we observed a subset of distinct T⁺ cells that expressed neither the amnion marker TFAP2A nor the PGC marker SOX17, but most of them co-expressed epiblast marker OCT4 (Fig. 4g and Supplementary Fig. 4d), suggesting a possible PSLI identity. In one of the embryos, we observed few T⁺ cells co-expressing GATA6, a marker of both PS and ExM, but no KDR⁺ ExMs were found (Fig. 4g and Supplementary Fig. 4d); in the remaining embryo, we did not observe T⁺ cells co-expressing GATA6, but found few T⁺ cells co-expressing KDR (Fig. 4g and Supplementary Fig. 4d), which may represent potential ExM precursors. These results display that potential PSLIs, detectable by IF staining, are specified in cultured human embryos at E11, yet no distinct ExM formation is observed at this stage.

Collectively, the scRNA-seq and IF staining results show that in cultured human embryos[3,26], potential PSLIs are specified prior to ExM formation (Fig. 4h). We previously identified the presence of PS cells (PS1 and PS2) as early as E11 in cultured human embryos using scRNA-seq[3]. Consistently, a recent study also detected T⁺ PS cells in cultured human embryos at E12 via IF staining[27]. These so-called PS cells[3,27]

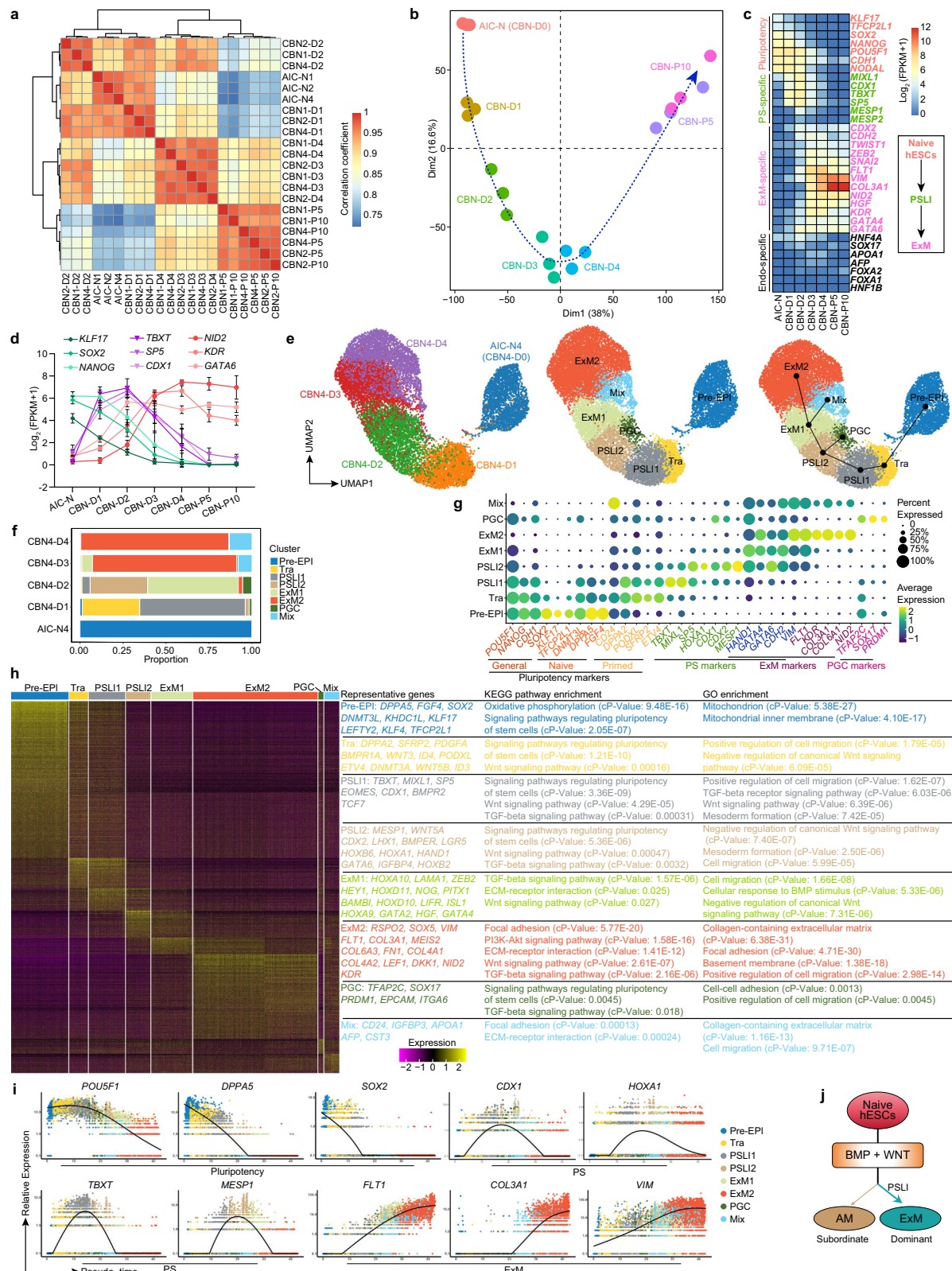

appear earlier than the acknowledged onset of gastrulation, thus their true identity–whether as PS cells or PSLIs–remains an open question.

### Primed and naive hESCs exhibit distinct responses to CB conditions

To investigate whether primed hESCs also rapidly produce ExMs in response to WNT and BMP signals like naive hESCs, we propagated primed hESCs in AIC medium (unless otherwise noted), known to support efficient derivation and expansion of hESCs via single-cell passaging[28], and then transferred them into CB conditions (Fig. 5a). After 4 days, these CB-treated primed hESCs (CBPs) rapidly lost colony morphology and OCT4 expression, but surprisingly, only a small number of CBPs expressed the ExM markers GATA6, FLT1, KDR, and SNAIL, and the vast majority of CBPs were positive for AM markers

**Fig. 3 | ExM specification from naive hESCs undergoes a primitive streak-like intermediate. a** Spearman correlation analysis of gene expression patterns in differentiation time courses for CBNs. **b** PCA of bulk RNA-seq data from differentiation time courses for CBNs, computed using the genes with FPKM ≥ 1 in at least one sample. **c** Heatmap of representative pluripotency and lineage marker genes in differentiation time courses for CBNs (left). Temporal sequence schematic of appearance of primitive streak-like intermediate (PSLI) and ExM (right). **d** Expression dynamics of the indicated marker genes in differentiation time courses for CBNs. $n = 3$ hESC lines; data are presented as mean ± SD. **e** UMAP visualization of scRNA-seq data from differentiation time courses for CBN4. UMAP plot is color-coded according to different time points or cell cluster identity annotations. Left, differentiation time courses; middle, clustering analysis; right, differentiation trajectory inferred by Slingshot. Pre-EPI, pre-implantation epiblast; Tra, transitional cell population; PGC, primordial germ cell; Mix, a mixture of

different types of cells. **f** Proportions of the indicated subtypes of cells in differentiation time courses for CBNs according to the results of scRNA-seq data. **g** Dot plots of candidate genes specific for cell subtypes. **h** Heatmap of differentially expressed genes among different cell subtypes. Representative genes (left), KEGG (middle) and GO (right) enrichment analysis are shown. $p$-adj of two-sided Wilcoxon's rank-sum test <0.05, $\log_2$ (FC) > 0.25 and expressed in > 25% of cells of the given cluster. Two-sided hypergeometric test for enrichment analysis, $p$-values were adjusted using the Benjamini-Hochberg method. cP-Value, corrected $p$-value. **i** Dynamic expression of *p*luripotency, PS, and ExM marker genes in differentiation time courses for CBNs over pseudotime. Each black line indicates the fitted expression trend of a gene over pseudotime. **j** Exogenous BMP and WNT signaling predominantly direct naive hESCs to an ExM fate through a PSLI, with few cells proceeding to AM fate. Source data are provided as a Source Data file. See also Supplementary Fig. 3.

TFAP2C, TFAP2A, and CK7, but negative for trophoblast (TrB) marker HAVCR1, endoderm markers SOX17 and FOXA1, and PS markers T, suggesting their AM identity (Fig. 5a and Supplementary Fig. 5a). PCA and clustering showed mutually exclusive groups of D4 CBNs and D4 CBPs (Fig. 5b, c), revealing a distinct difference in cell identity. Gene expression analysis displayed that, compared with D4 CBNs, D4 CBPs significantly downregulated ExM marker genes and upregulated AM marker genes (Fig. 5d and Supplementary Fig. 5b). The scRNA-seq results revealed that D4 CBPs consisted of 3 clusters: the dominant AM1 (76.4%) and the subordinate AM2 (4.8%) and ExM (18.8%) (Supplementary Figs. 1e, 5c). Expressions of *LUM*, *WNT6*, *IGFBP3*, *GATA3*, *TFAP2A*, and *KRT7* were all upregulated in both AM1 and AM2 clusters, along with *GABRP* and *EPAS1* mainly in AM1, and *TP63* and *ITGA6* mainly in AM2 (Supplementary Fig. 5c–e). Clustering and RNA velocity map further showed that AM1 and AM2 had similar identities and that AM2 appeared to be specified from AM1 (Supplementary Fig. 5f, g). The ExM cluster, which did not express AM marker genes, showed a different identity from AM1/AM2 and upregulated expressions of *GATA6*, *KDR*, *GATA4*, *VIM*, *FLT1*, and *COL3A1* (Supplementary Fig. 5c–f), indicating its ExM identity.

To further compare naive and primed hESC derivatives grown under CB conditions, we collected CBNs and CBPs across various time points. Bulk RNA-seq data showed that CBNs and CBPs followed two distinct developmental trajectories (Fig. 5e and Supplementary Fig. 5h). In contrast to CBNs, CBPs displayed a modest increase in ExM-related genes, but a significant upregulation of AM-related genes during the developmental progression, alongside an absence of endodermal gene expression (Fig. 5f and Supplementary Fig. 5i). Interestingly, CBPs also showed an initial increase followed by a decrease in expression of PS-related genes, albeit lower levels and faster declines than in CBNs (Supplementary Fig. 5i, j), suggesting the presence of PSLIs in CBPs. We integrated scRNA-seq datasets from D0-D4 CBPs (Fig. 5g, Supplementary Figs. 1e, 3c), based on specific marker genes, 6 major clusters were identified, including PostL-EPI, transitional cell population 1 (Tra1), transitional cell population 2 (Tra2), AM, PSLI, and ExM (Fig. 5g–i). PostL-EPI and Tra1 corresponded almost exactly to primed hESCs (OCT4⁺SOX2⁺) and D1 CBPs, respectively (Fig. 5g–j). A small number of Tra1 upregulated expression of PS genes (*EOMES*, *MIXL1*, *SP5*, and *TBXT*), and most of Tra1 upregulated expression of initial AM genes (*TFAP2A* and *GATA3*), accompanied by complete *SOX2* depletion (Fig. 5i, j). Tra2 further upregulated expression of other AM genes (*WNT6* and *KRT7*), while PSLI upregulated expressions of PS marker *MESP1* and PS/ExM markers *VIM* and *CDH2*, accompanied by downregulation of *EOMES*, *MIXL1*, and *SP5* (Fig. 5i, j). Compared with other clusters, AM and ExM expressed the highest levels of AM- and ExM-related genes, respectively (Fig. 5i). Importantly, trajectory analysis and gene expression dynamics revealed that ExMs in CBPs originated from PSLIs (Fig. 5g, j, k), consistent with the bulk RNA-seq results (Supplementary Fig. 5i, j) and similar to ExM specification in CBNs (Fig. 3c–e and Supplementary Fig. 3d). Integration and

correlation analysis of scRNA-seq datasets between CBNs and CBPs across different time course showed a similar progression of ExM specification, namely from pluripotency to the PSLI and ultimately to the ExM (Fig. 5l, m).

To assess the response of primed hESCs grown under alternative medium to CB conditions, we transferred mTeSR- and E8-cultured hESCs into CB conditions. Interestingly, mTeSR-hESCs showed a phenotype comparable to AIC-hESCs under CB conditions, whereas E8-hESCs yielded significantly more ExMs (Fig. 5n and Supplementary Fig. 5a, k). Because the basal media of AIC and mTeSR are similar but differ significantly from that of E8[28], this may suggest that the composition of the basal medium influence the response of hESCs to CB conditions. However, E8-hESC derivatives also contained a significant proportion (approximately half) of TFAP2A⁺TFAP2C⁺CK7⁺HAVCR1⁻ AM cells (Fig. 5n and Supplementary Fig. 5k), which was distinctly different from naive hESCs (Fig. 5n and Supplementary Fig. 1a, k–m). To determine whether WNT signaling also plays a switching role in the fate specification of primed (AIC and E8) hESCs, we replaced CHIR with IWP2 in the presence of BMP4, and observed that almost all primed hESCs differentiated into TFAP2C⁺CK7⁺TFAP2A⁺HAVCR1⁻ AM cells on D4, accompanied by depletion of GATA6⁺ ExMs (Supplementary Fig. 5l), revealing a similar effect of WNT signaling on primed and naive hESCs in the context of BMP activation (Figs. 2l, 5o).

Collectively, these results indicate that primed hESCs yield a significant population of AM cells in CB conditions, unlike naive hESCs which mainly differentiate into ExMs, showing that the initial pluripotent state plays a essential role in cell fate specification. However, it is consistent that both primed and naive hESC-derived ExMs are primarily specified from PSLIs and depend on WNT signaling (Figs. 2l, 3j, 5o).

## Primed hESCs efficiently differentiate into ExMs in response to CBA conditions

Because ExMs derived from hESCs undergoes a PSLI, and primed hESCs differentiate largely into AM cells under CB conditions, we hypothesized that promoting a PS-like phenotype and inhibiting the AM fate could enhance the differentiation efficiency of primed hESCs into ExMs. Previous studies have shown that Activin/Nodal signaling promotes PS formation[23,29] and inhibits AM specification[30] in primate ESCs. We therefore speculated that the exogenous addition of Activin-A (an activator of Activin/Nodal signaling) in CB conditions might promote the differentiation of primed hESCs into ExMs at the expense of AM specification. As expected, these CB+Activin-A-treated primed hESCs (CBAPs) were converted to mesenchymal cells, and most of them expressed GATA6, KDR, and VIM, accompanied by a small number of residual OCT4⁺E-Cad⁺ cells on D4 (Fig. 6a and Supplementary Fig. 6a). By day 5, the expression of pluripotency markers was depleted in CBAPs (Supplementary Fig. 6b). In contrast to CBPs, CBAPs exhibited a distinct and homogeneous mesenchymal morphology (Fig. 6b). Of note, very few ( ~1%) TFAP2A⁺ cells were observed and

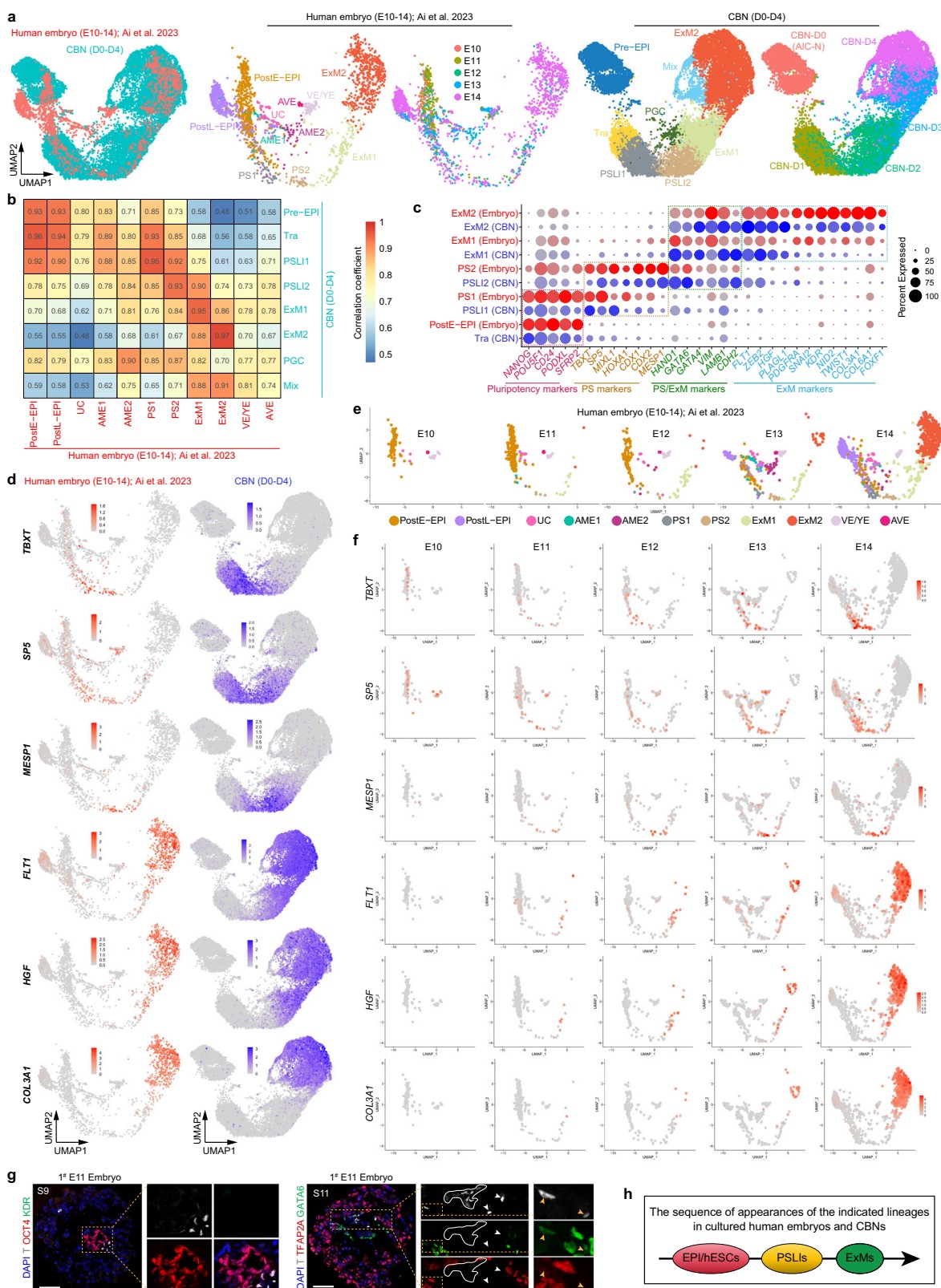

more than 90% of CBAPs expressed GATA6, along with the presence of KDR and VIM (Figs. 6c–6e and Supplementary Fig. 6b). Obviously, PCA and clustering showed distinct developmental trajectories for CBAPs and CBPs, as well as their mutually exclusive groups (Fig. 6f and Supplementary Fig. 6c). Gene expression analysis revealed that primed hESCs upregulated the markers of ExM but not AM under CBA conditions (Fig. 6g and Supplementary Fig. 6d), and the resulting CBAPs

showed similar GO and KEGG enrichment as CBNs when compared with their initial hESCs (Fig. 1e and Supplementary Fig. 6e). Consistently, the scRNA-seq data showed that few AM (~1%) and Endo (endoderm, ~1%) cells were detected, while the remaining cells exhibited ExM characteristics and were composed of ExM1 and ExM2 subgroups (Fig. 6h, i, Supplementary Fig. 6f, g). Furthermore, D5 CBAPs were stably propagated over 10 passages and maintained

**Fig. 4 | Similar developmental dynamics of ExMs between cultured human embryos and CBNs. a** UMAP visualization and plots of integration analysis of scRNA-seq data from cultured human embryos from embryonic day (E10–14)[3] and differentiation time courses for CBN4. **b** Correlation analysis of different clusters in cultured human embryos and CBNs. **c** Dot plots of candidate genes specific for the indicated cell subtypes in cultured human embryos and differentiation time courses for CBN4. **d** UMAP plots of the indicated genes expressed in cultured human embryos and differentiation time courses for CBN4. **e** Split UMAP visualization of scRNA-seq data from cultured human embryos according to different developmental time points. **f** UMAP plots showing the dynamics of the indicated

genes expressed in cultured human embryos. **g** IF staining of the indicated markers in the 1[#] extended cultured human embryo at E11. White numbers indicate section (S) numbers, white solid circles and white arrowheads indicate T⁺GATA6⁻TFAP2A⁻ PSLIs, yellow boxes and yellow arrowheads indicate few T⁺GATA6⁺TFAP2A⁻ PSLIs. We cultured four E6 blastocysts to E11 and observed T⁺ PSLIs in two embryos (1[#] and 2[#], see also Supplementary Fig. 4d) but did not detect ExMs in any of the four embryos. **h** Schematic showing the sequence of appearances of PSLIs and ExMs. Scale bars, 100 μm. Source data are provided as a Source Data file. See also Supplementary Fig. 4.

characteristics of expandable ExMs at different passages in hTS-M (Fig. 6f, j, Supplementary Fig. 6c, h).

To explore the developmental progression, we checked the gene expression dynamics of CBAPs. Similar to CBNs (Fig. 3c, d), we observed that PS-related genes such as *SP5*, *MIXL1*, *TBXT*, *MESP1/2*, *CDX1*, and *GATA6* were significantly upregulated on D1 and gradually downregulated after peaking on D2, with stable *GATA6* (also ExM marker) expression level after D3 (Fig. 6g and Supplementary Fig. 6d). At the same time, the ExM-related genes were continuously upregulated and reached stable expression levels on D4/5 (Fig. 6g and Supplementary Fig. 6d). IF staining further displayed that the proportion of PS marker T⁺ cells was 74.6%, 88.2%, and 94.9% on D1, D2, and D3, respectively, and then sharply depleted on D4, accompanied by a continuously increasing proportion of GATA6⁺, KDR⁺, and FLT1⁺ cells to over 90% after D4 (Fig. 6k, Supplementary Fig. 6i, j). To further validate the developmental dynamics of CBAPs, we employed the MESP1-mTomato knock-in reporter hESC line[31] to track the time course of the PS marker MESP1. Live-cell imaging showed that MESP1-mTomato⁺ cells appeared at 29 h, and their proportion progressively increased and reached a peak (~87.6%) at 64 h, then declined and depleted at 104 h (Fig. 6l, Supplementary Fig. 6k, Supplementary Movies 1, 2). We sorted over 90% of MESP1-mTomato⁺ cells at 72 h and cultured them under CBA conditions for an additional 48 h. These cells expressed KDR, VIM, GATA6, and FLT1, providing direct evidence that MESP1⁺ PSLIs differentiate into ExMs (Fig. 6m–o).

Similar to AIC-hESCs, E8-hESCs also efficiently differentiated into ExMs under CBA conditions (Fig. 6p). As expected, the addition of exogenous Activin-A to IB conditions almost completely abolished the AM specification of primed hESCs (Supplementary Figs. 5l, 6l). Together, the exogenous addition of Activin-A in CB conditions inhibits the AM fate of primed hESCs and promotes efficient ExM specification, predominantly through a PSLI (Fig. 6q).

### Naive hESCs efficiently differentiate into ExMs in response to CBA conditions

To determine if naive hESCs also efficiently differentiate into ExMs in CBA conditions, we subjected AIC-N hESCs to CBA medium for 5 days. These CBA-treated AIC-N hESCs (CBANs) acquired a mesenchymal morphology, and almost all cells were positive for GATA6 and KDR, but lacked expression of TFAP2A and CK7, with the presence of a small number of SOX17⁺ cells and some residual OCT4^weak cells (Fig. 7a–c). Consistent with CBAPs, bulk RNA-seq data showed that CBANs significantly upregulated the expression of ExM but not AM marker genes after D3, and revealed a differentiation progression of CBANs from pluripotency to PSLI, and then to ExM (Fig. 7d and Supplementary Fig. 7a); but differently, CBANs demonstrated a delayed initiation of PS genes and an extended exit of pluripotency/PS genes compared to CBAPs (Supplementary Fig. 7b). We next identified the cell type composition of D5 CBANs via scRNA-seq data, and annotated 3 subpopulations consisting of Endo (8.6%), ExM (82.7%), and haematoendothelial progenitor (HEP, 8.7%) (Fig. 7e and Supplementary Fig. 7c). Endo and ExM/HEP highly expressed endoderm (*FOXA2*, *SOX17*, *APOA1*, and *GSC*) and ExM (*FLT1*, *VIM*, *COL3A1*, and *KDR*) marker genes, respectively (Fig. 7f and Supplementary Fig. 7c). Consistent with the

staining and bulk RNA-seq results, the scRNA-seq data showed that AM cells were missing from CBANs (Fig. 7c–f, Supplementary Fig. 7a, c). In contrast to D5 CBAPs, a group of HEPs expressing hematopoietic markers (*KDR*, *PECAM1*, *HHEX*, *CD34*, *MEF2C*, *SOX17*, and *CDH5*) was specified in D5 CBANs, which was further confirmed by IF staining for CD31⁺KDR⁺ and CD34⁺KDR⁺ cells (Fig. 7e–g, Supplementary Fig. 7c, d). When inoculated into hTS-M, D5 CBANs could be stably propagated for more than 10 passages and exhibited characteristics of expandable ExMs (Supplementary Fig. 7e–g), similar to D4 CBNs and D5 CBAPs. To validate that naive hESCs-derived ExMs undergo PSLIs (Fig. 7d and Supplementary Fig. 7a), we isolated MESP1-mTomato⁺ cells at 72 h and cultured them for an additional 48 h. These cells expressed KDR, VIM, GATA6, and FLT1, indicating that MESP1⁺ PSLIs progressed into ExMs (Fig. 7h–k). Consistent with gene expression (Supplementary Fig. 7b), flow cytometry results indicated a delayed expression of the PS marker MESP1 in CBANs compared to CABPs (Figs. 6n, 7i). As expected, naive hESCs grown in PXGL and AIC-N media showed consistent responses to CBA conditions (Fig. 7c, g, Supplementary Fig. 7h). Similar to primed hESCs, the addition of Activin-A under IB conditions almost completely eliminated AM specification in naive hESCs (Fig. 2g, h, j, Supplementary Figs. 2c–e, 7i).

These data indicate that, similar to primed hESCs, the addition of Activin-A inhibits the AM fate of naive hESCs while promoting ExM specification, largely through a PSLI (Fig. 7l). However, compared to primed hESCs, naive hESCs exhibit a delayed initiation of PS genes and an extended exit of pluripotency/PS marker genes, along with a loss of AM lineages and the acquisition of a small number of hematopoietic cells.

### Naive and primed hESCs show both shared and distinct aspects in ExM specification

We further examined the developmental trajectories of naive and primed hESCs under CB and CBA conditions, and found that the developmental progression of naive hESCs was highly similar and synchronized under both conditions (Fig. 8a). In contrast, primed hESCs exhibited two distinct developmental branches: the CBAP branch gradually converged towards, whereas the CBP branch increasingly diverged from the developmental routes of naive hESCs (Fig. 8a). The extended culture in hTS-M enabled CBAPs and CBANs/CBNs to progress to a stable state at P5/10, with a more similar identity than in the early stages (D4/5) (Fig. 8a). Consistent with the PCA result (Fig. 8a), correlation analysis further confirmed that early (D3-D5) and late (P5/P10) CBAPs clustered together with CBNs and CBANs at the same stage but were distinct from CBPs (Supplementary Fig. 8a), suggesting similar cell identity and developmental processes among CBAPs, CBNs and CBANs. However, CBANs and CBNs were always closer together than CBAPs in both early and late stages, indicating the influence of the pluripotent state on cell characteristics (Fig. 8a and Supplementary Fig. 8a). To further characterize CBNs, CBANs, and CBAPs, we compared their differentially expressed genes at different stages. Consistent with PCA and correlation results (Fig. 8a and Supplementary Fig. 8a), differential expression analysis showed that CBNs and CBAPs exhibited the largest difference, followed by CBANs and CBAPs, with CBANs and CBNs indicating the smallest difference

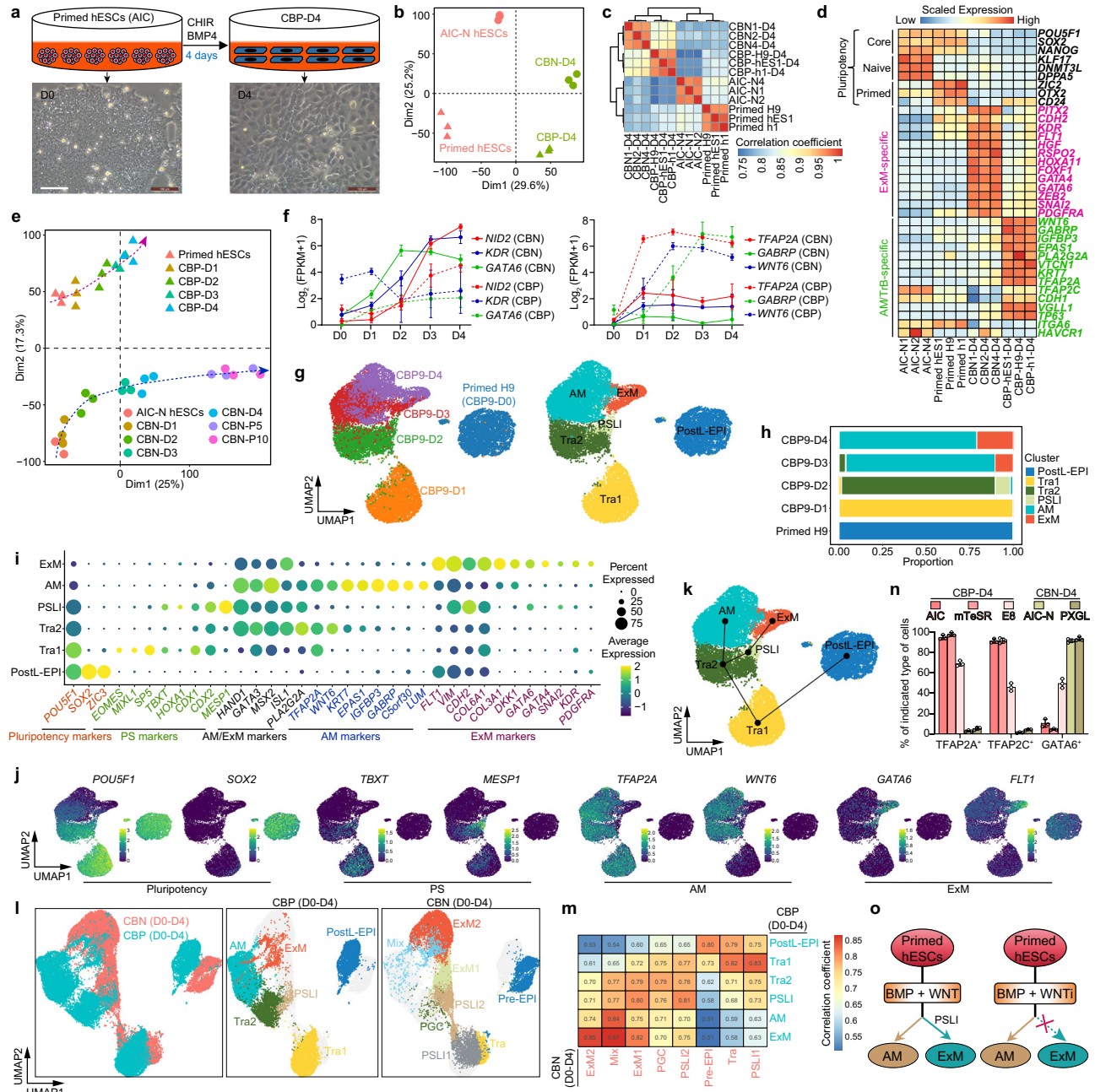

**Fig. 5 | Primed hESCs largely commit to an AM fate under CB conditions.**
**a** Schematic and representative brightfield images illustrating primed hESCs and their derivatives induced after 4 days under CB conditions. CBP, CB-treated primed hESCs. Scale bars, 100 μm. **b** PCA of bulk RNA-seq data from naive/primed hESCs and their derivatives induced after 4 days under CB conditions, computed using the genes with FPKM ≥ 1 in at least one sample. **c** Spearman correlation analysis of gene expression patterns in naive/primed hESCs and their derivatives induced after 4 days under CB conditions. **d** Heatmap of representative pluripotency, ExM, and AM/TrB marker genes in naive/primed hESCs and their derivatives induced after 4 days under CB conditions. Values represent log₂ (FPKM + 1) scaled by gene expression across samples. **e** PCA of the gene expression profiles of differentiation time courses for CBPs and CBNs, computed using the genes with FPKM ≥ 1 in at least one sample. Primed and naive hESCs undergo two distinct developmental trajectories. **f** Expression dynamics of the indicated marker genes in differentiation time courses for CBNs and CBPs. *n* = 3 hESC lines; data are presented as mean ± SD. **g** UMAP visualization of scRNA-seq data from differentiation time courses for CB-

treated primed H9 (CBP9). UMAP plot is color-coded according to different time points (left) or cell cluster identity annotations (right). PostL-EPI, post-implantation late epiblast; Tra1/2, transitional cell population 1/2. **h** Proportions of the indicated subtypes of cells in differentiation time courses for CBP9 according to the scRNA-seq data. **i** Dot plots of candidate genes specific for cell subtypes. **j** UMAP plots of the indicated genes expressed in differentiation time courses for CBP9.
**k** Differentiation trajectory inferred by Slingshot. **l** UMAP visualization and plots of integration analysis of scRNA-seq data from differentiation time courses for CBN4 and CBP9. **m** Correlation analysis of different clusters in differentiation time courses for CBN4 and CBP9. **n** Proportion of the indicated cell types in D4 CBPs/CBNs from primed/naive hESCs cultured in different media. *n* = 3 independent experiments; data are presented as mean ± SD. **o** Schematics of signaling principles in the specification of ExM and AM lineages from primed hESCs. WNTi, WNT inhibition. Source data are provided as a Source Data file. See also Supplementary Fig. 5.

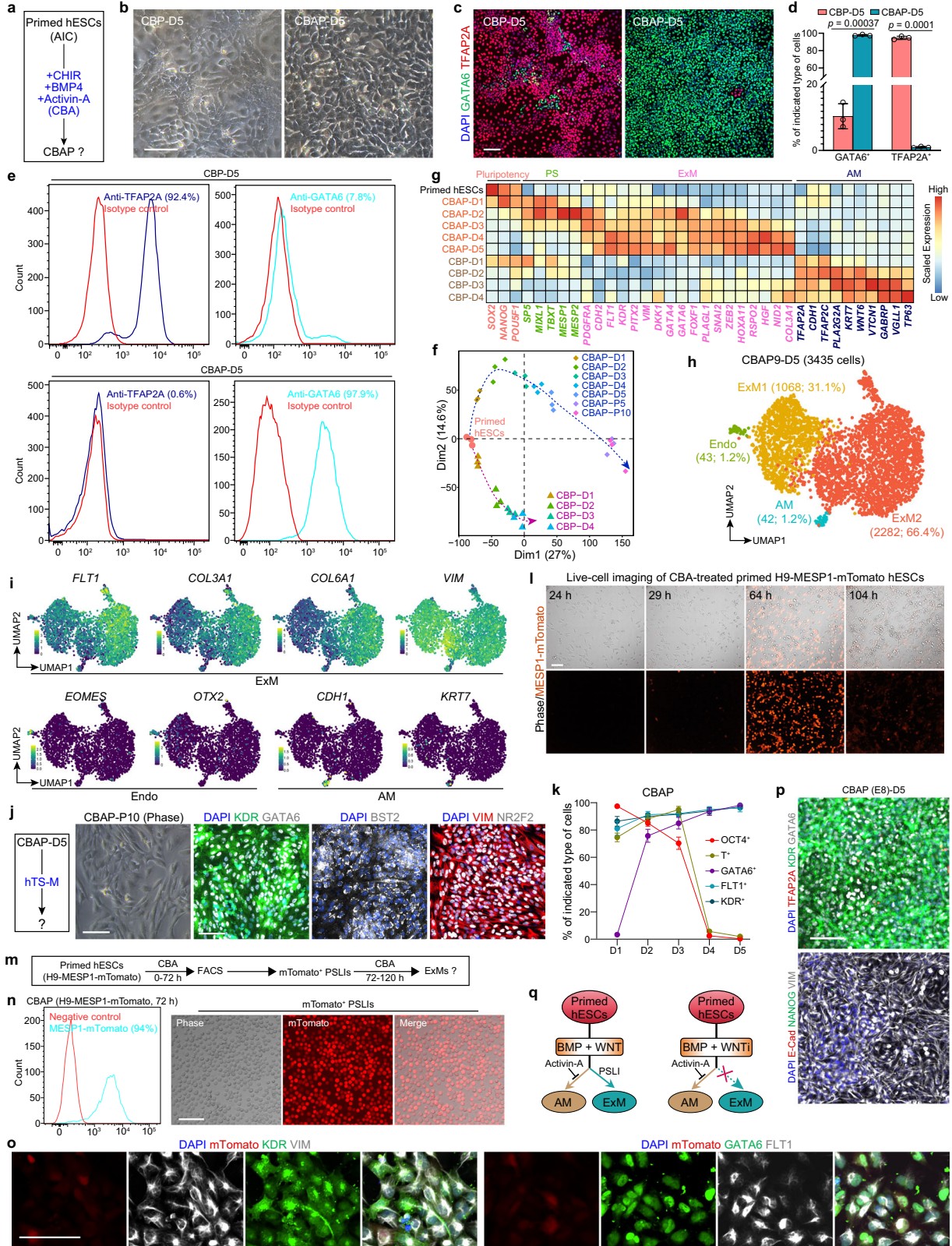

(Supplementary Fig. 8b). Notably, the protocadherin (PCDH) genes, which play a crucial role in cell adhesion, synaptic connections, and signal transduction, were upregulated in CBAPs. In contrast, the metallothionein (MT) genes, known for their antioxidant and anti-inflammatory effects, were upregulated in both CBNs and CBANs (Supplementary Fig. 8b). The endocrine-imprinted *IGF2* and *DLK1* genes are expressed in the placenta and are required for maternal

metabolic adaptations to pregnancy; a loss or low level of their expression are associated with fetal growth restriction[32–35]. Interestingly, we observed high expression of *IGF2* and *DLK1* in CBAP, rather than in CBN and CBAN (Fig. 8b and Supplementary Fig. 8b). Similarly, *IGF2* and *DLK1* were also specifically expressed in the ExMs of gastrulating embryos[4], but were absent in both the ExMs of cultured pre-gastrulating embryos[3] and naive hESCs-derived ExMs[12] (Fig. 8c). These

**Fig. 6 | Exogenous addition of Activin-A switches the fate of primed hESCs from AM to ExM. a** Schematic of the treatment of primed hESCs with a combination of CHIR, BMP4, and Activin-A (CBA). **b** Representative brightfield images illustrating CBP-D5 and CBA-treated primed hESCs on day 5 (CBAP-D5). **c** IF staining for TFAP2A and GATA6 in CBP-D5 and CBAP-D5. **d** Proportion of the indicated cell types in CBP-D5 and CBAP-D5. Unpaired *t*-test with Welch correction, *n* = 3 independent experiments; data are presented as mean ± SD. **e** Flow cytometry analysis of TFAP2A and GATA6 expression in CBP-D5 and CBAP-D5. **f** PCA of the gene expression profiles of differentiation time courses for CBPs and CBAPs, computed using the genes with FPKM ≥ 1 in at least one sample. Primed hESCs develop along two distinct trajectories under CB and CBA conditions, respectively. **g** Heatmap of representative pluripotency, PS, ExM, and AM marker genes in primed hESCs and their derivatives, including the differentiation time courses for both CBPs and CBAPs. Values represent log$_2$ (FPKM + 1) scaled by gene expression across samples. **h** UMAP visualization of single-cell transcriptomics from CBA-treated primed H9 hESCs on day 5 (CBAP9-D5). UMAP plot is color-coded according to cell cluster identity annotations. **i** UMAP plots of the indicated genes expressed in CBAP9-D5. **j** Schematic of CBAP-D5 inoculated into hTS-M for extended culture (left), and representative brightfield and IF staining images for the indicated markers in expandable CBAPs at passage 10 (right). **k** Proportion of the indicated cell types in Supplementary Fig. 6i, j; *n* = 3 independent experiments; data are presented as mean ± SD. **l** Live-cell imaging showing the expression dynamics of mTomato in the differentiation time courses for MESP1-mTomato knock-in reporter CBAPs (see also Supplementary Movies 1, 2). **m** Schematic of fluorescence-activated cell sorting (FACS) for MESP1-mTomato⁺ PSLIs. **n** Flow cytometry analysis of MESP1-mTomato expression in CBAPs at 72 hours (left), and brightfield and fluorescence images of sorted PSLIs (right). **o** IF staining of the indicated markers in sorted PSLIs after 48 hours of additional culture under CBA conditions. **p** IF staining for the indicated markers in D5 CBAPs, primed hESCs were cultured in E8 medium. **q** Schematic of signaling principles and developmental dynamics in the specification of ExM and AM lineages from primed hESCs. Scale bars, 100 μm. Source data are provided as a Source Data file. See also Supplementary Fig. 6.

results indicate that, in terms of *IGF2* and *DLK1* expression, ExMs derived from primed hESCs are similar to gastrulating ExMs, while ExMs from naive cells resemble pre-gastrulation ExMs. To rule out the effects of different culture conditions, we focused on the differences between CBAPs and CBANs. We identified 541 and 423 genes upregulated and 513 and 491 genes downregulated in CBAPs relative to CBANs in early (D5) and late (P5) stages, respectively (Supplementary Fig. 8b). The top 25 CBAP and CBAN marker genes and differentially expressed transcription factors were presented (Fig. 8d and Supplementary Fig. 8c). GO terms and KEGG enrichment highlighted homophilic cell adhesion, extracellular matrix, growth factor activity, and TGFβ/ Hippo/MAPK signaling pathway in D5 CBAPs, contrasting with cellular response to zinc ion, extracellular matrix organization, inflammatory response, mineral absorption, and TNF/Jak−STAT signaling pathway in D5 CBANs (Supplementary Fig. 8d). PI3K−Akt signaling pathway was included in both D5 CBAPs and D5 CBANs, but which were enriched with different genes (Supplementary Fig. 8d). In the late stage (P5), CBAPs were enriched with genes related to the homophilic cell adhesion, calcium ion binding, extracellular matrix, glycosphingolipid biosynthesis, 2−Oxocarboxylic acid metabolism, pantothenate and CoA biosynthesis, aldosterone synthesis and secretion, and phosphatidylinositol signaling system (Supplementary Fig. 8e); by contrast, CBANs were enriched with genes for cytokine−mediated signaling pathway, lipid catabolic process, extracellular matrix organization, oxidation−reduction process, arachidonic acid metabolism, ovarian steroidogenesis, taurine and hypotaurine metabolism, and glutathione metabolism (Supplementary Fig. 8e). These data reveal that the ExMs derived from naive and primed hESCs display disparities in various aspects, including cell adhesion and extracellular matrix, the activities of growth factors and cytokines, signaling pathway regulation, as well as biosynthetic and metabolic processes.

To compare the ExMs derived from different conditions and sources, we integrated scRNA-seq datasets from D4 CBNs, D4 CBPs, D5 CBANs, D5 CBAPs, human CS7 gastrulating embryo[4], cultured human peri-implantation embryos[3], and naive hESC derivatives in hTS-M[12]. The integration and correlation analysis showed that the ExMs from this study, regardless of being derived from naive or primed hESCs, were largely similar to ExMs in human embryos[3,4] and early ExMs from naive hESCs[12] (Fig. 8e and Supplementary Fig. 8f), suggesting their shared ExM identity. We further integrated scRNA-seq datasets from D5 CBAP and D5 CBAN and observed mutually exclusive clustering in ExMs from different sources with little to no overlap (Fig. 8f). Three distinct ExM clusters, namely ExM from D5 CBAN and ExM1/ExM2 from D5 CBAP, exhibited markedly different gene expression patterns (Fig. 8g). We performed co-expression network analysis using hdWGCNA[36] and identified 11 co-expression modules within three ExM clusters (M1−M11, Fig. 8h). We found that M1 and M3 are specific to

ExM and ExM1/2, respectively (Fig. 8h and Supplementary Fig. 8g). GO enrichment analysis revealed M1 to be associated with translation functions such as regulation of translation, mitochondrial translation, and ribosome biogenesis. In contrast, M3 was enriched in cytoskeletal regulation, including axon guidance, regulation of dendrite development, and actin filament organization (Fig. 8i). These results suggest that ExM from D5 CBAN may have a stronger ability to synthesize proteins (such as collagen), while ExM1/2 from D5 CBAP may have enhanced signal transmission and migration capabilities.

Collectively, both primed and naive hESCs can efficiently differentiate into ExMs under CBA conditions, yet the initial pluripotent state influences the transcriptional characteristics of ExMs from hESCs (Fig. 8j).

## Discussion

Although previous reports have derived ExMs from naive[12] and primed[13] hESCs, the protocols are inefficient and/or time-consuming, leading to a poor understanding of the developmental process and signaling principles of human ExM specification. By addressing these barriers to ExM differentiation, this study explored the developmental dynamics and signaling principles of human ExM specification. We found that combined activation of BMP, WNT, and Activin/Nodal signaling pathways rapidly (within 4-5 days) and efficiently ( ~90%) drives ExM specification from both naive and primed hESCs (Fig. 7l). The methodology we established for inducing ExM generation provides an important foundation for both basic research and translational applications of human ExMs. Notably, human ExM specification underwent a developmental progression from pluripotency to PSLIs and then to ExMs, regardless of the pluripotent phase of the initial hESCs (Fig. 7l). In 3D-cultured human embryos, we further observed the formation of PSLIs prior to the specification of ExMs (Fig. 4h). Although additional origins of ExMs, such as the hypoblast, are possible, our findings open up the possibility that PSLIs in the embryonic disc also contribute to the origin of the earliest ExMs, potentially establishing a conservation in ExM specification between humans and rodents[37,38].

Naive and primed hESCs are the in vitro counterparts of Pre-EPI and PostL-EPI in human embryos, respectively (Fig. 8j)[15,16]. We found that dual activation of BMP and WNT signaling drove the rapid and efficient specification of naive hESCs to ExMs (Fig. 3j), but under the same conditions, primed hESCs largely progressed to AM fate (Fig. 5o), indicating a significant difference in response to BMP and WNT signaling between primed and naive hESCs. Although naive hESCs exhibited a transient upregulation of primed genes such as *CD24*, *DPPA2*, and *ETV4* under CB conditions on D1, this was accompanied by the loss of pluripotency marker *SOX2* and an upregulation of PS genes, including *TBXT*, *MIXL1*, and *SP5* (Fig. 3e, g). A previous study has reported that the transition of naive hESCs to a stable primed state

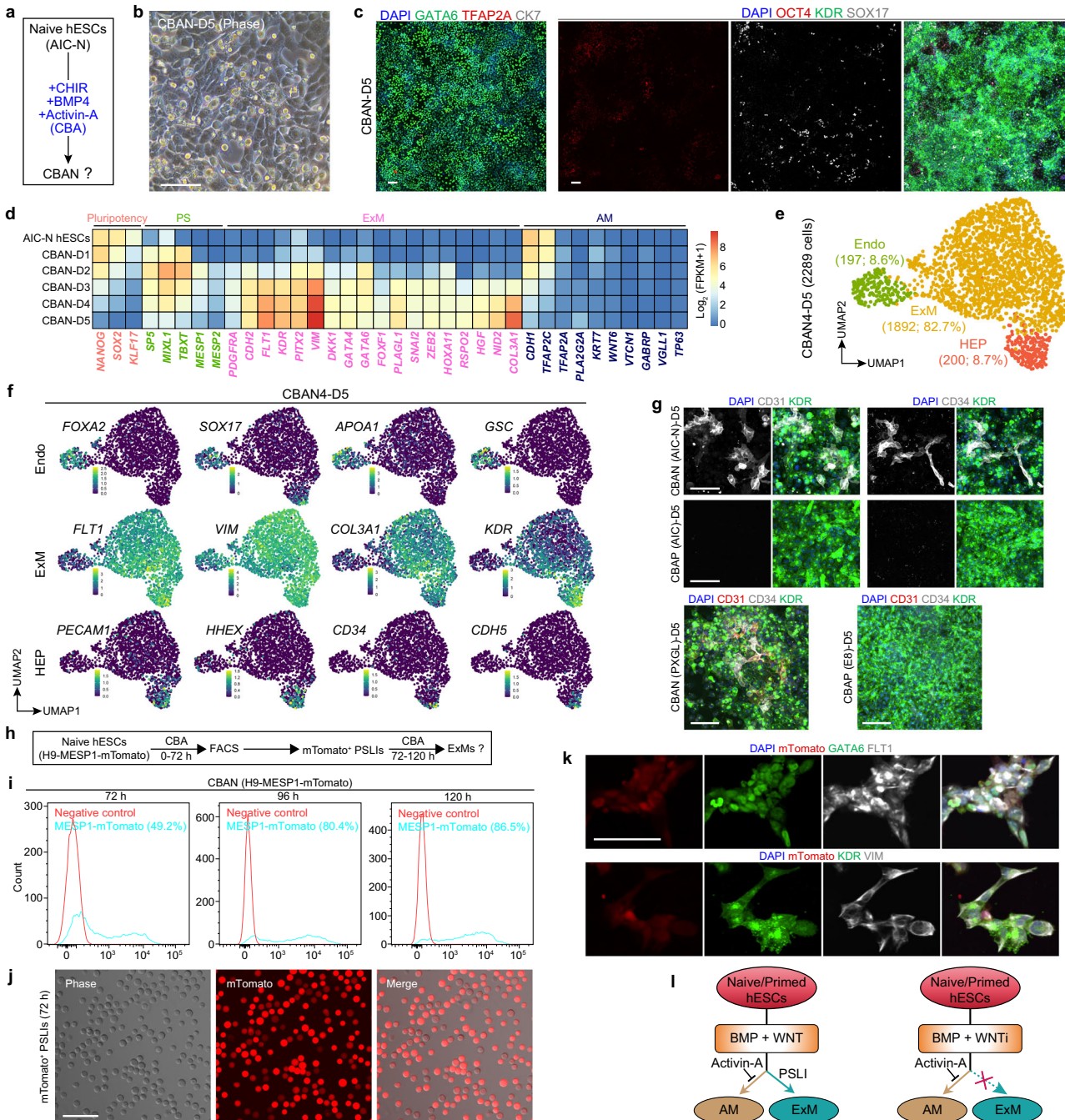

**Fig. 7 | Naive and primed hESCs exhibit both similarities and differences in their response to CBA conditions. a** Schematic of CBA treatment of naive hESCs. **b** Brightfield image illustrating CBA-treated naive hESCs on day 5 (CBAN-D5). **c** IF staining images for the indicated markers in CBAN-D5. **d** Heatmap illustrating the expression of representative pluripotency, PS, ExM, and AM marker genes across the differentiation time courses of CBANs. **e** UMAP visualization of single-cell transcriptomics from CBA-treated AIC-N4 hESCs on day 5 (CBAN4-D5). UMAP plot is color-coded according to cell cluster identity annotations. HEP, haemato-endothelial progenitor. **f** UMAP plots of the indicated genes expressed in CBAN4-D5. **g** IF staining images for the hematopoietic markers KDR, CD31, and CD34 in

CBAN-D5 and CBAP-D5, naive and primed hESCs were cultured under the indicated conditions. **h** Schematic of FACS for MESP1-mTomato+ PSLIs. **i** Flow cytometry analysis of MESP1-mTomato expression across the differentiation time courses of CBANs. **j** Brightfield and fluorescence images of sorted PSLIs at 72 hours. **k** IF staining of the indicated markers in sorted PSLIs after 48 hours of additional culture under CBA conditions. **l** Schematics illustrating the signaling principles and developmental dynamics in ExM specification from naive and primed hESCs. Scale bars, 100 μm. Source data are provided as a Source Data file. See also Supplementary Fig. 7.

requires 10 days[19]. Consequently, naive hESCs rapidly initiated differentiation in CB conditions, progressing from naive pluripotency to PSLIs and then to ExMs, prior to transitioning to the primed state (Figs. 3e, g, 5l). These results indicate that naive hESCs, despite not responding to direct germ layer induction[19], can differentiate into ExMs in response to exogenous signals via PSLIs, similar to their

rapid differentiation into trophoblasts in response to MEK and TGF-β inhibitors[39,40]. The exogenous addition of Activin-A endowed primed and naive hESCs with compatibility for efficient ExM specification at the expense of AM differentiation (Fig. 7l). However, there were transcriptional differences between the two sources of ExMs (Fig. 8j), especially the differential expression of imprinted *IGF2* and *DLK1*

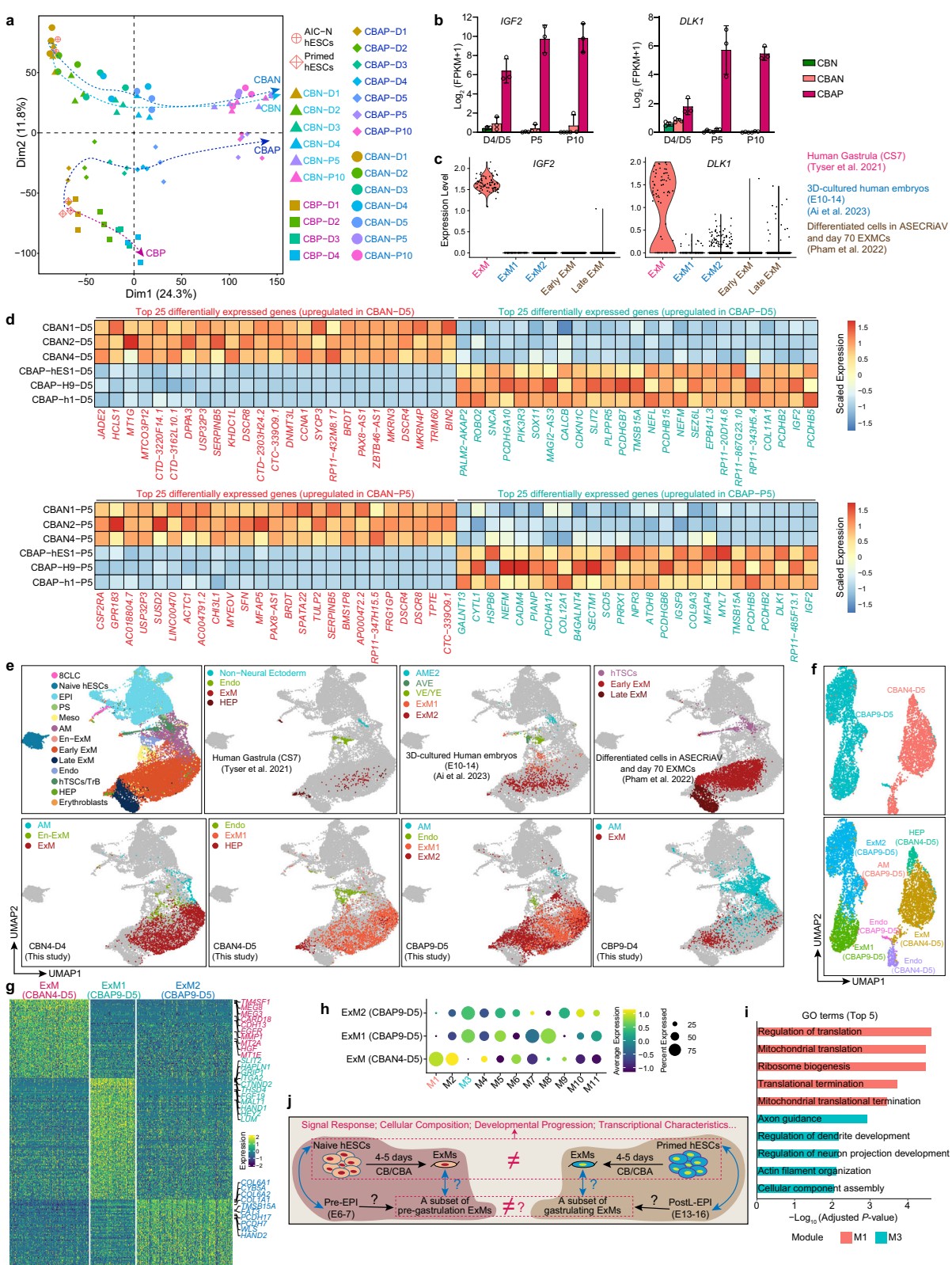

genes (Fig. 8b), suggesting that the two sources of ExMs may have different functions in embryonic development and maternal-fetal interaction[32–35]. The current view is that human ExM specification consists of two independent waves in primates, namely the early wave before gastrulation and the late wave during gastrulation (Fig. 8j)[1,14], but it is unknown whether there are differences in the specification principles and transcriptional characteristics of the two

waves of ExMs. Our results showed that the initial pluripotent state had an important impact on the specification of ExMs, which may imply that in human embryos: (1) ExM specification from epiblast is spatiotemporally specific in signaling response and developmental progression; and (2) there are transcriptional and functional differences in the ExMs derived from epiblast at different developmental stages (Fig. 8j).

**Fig. 8 | Naive and primed hESCs exhibit both similarities and differences in ExM specification. a** PCA analysis of gene expression profiles across differentiation time courses for CBNs, CBANs, CBPs, and CBAPs, computed using the genes with FPKM ≥ 1 in at least one sample. **b** Bar charts illustrating the expression of *IGF2* and *DLK1* across various samples. n = 3 hESC lines; data are presented as mean ± SD. **c** Violin plots of *IGF2* and *DLK1* expression in ExMs from various sources. **d** Heatmap of the top 25 differentially expressed genes between CBANs and CBAPs at different time points (D5/P5). Log₂ (FC) ≥ 1, padj < 0.05, and FPKM ≥ 3 in at least two samples. Values represent log₂ (FPKM + 1) scaled by gene expression across samples. Two-sided Wald test, *p*-values were adjusted using the Benjamini-Hochberg method. **e** UMAPs illustrating the integration of CBN4-D4, CBAN4-D5, CBP9-D4, and CBAP9-D5 with reference human embryonic cells and naive hESC derivatives in the hTS-M

(ASECRiAV). **f** UMAP visualization of scRNA-seq data from CBAN4-D5 and CBAP9-D5. UMAP plot is color-coded according to different sources (top) or cell cluster identity annotations (bottom). **g** Heatmap of differentially expressed genes among different cell subtypes. *p*-adj of two-sided Wilcoxon's rank-sum test < 0.05, log₂ (FC) > 0.5 and expressed in > 25% of cells of the given cluster. **h** Dot plots illustrating gene co-expression modules specific to cell subtypes. M, module. **i** Bar plots displaying the GO term enrichment results for M1 and M3. Two-sided Fisher's exact test, *p*-values were adjusted using the Benjamini-Hochberg method. **j** Schematic illustrating differences in ExM specification between naive and primed hESCs. Black arrows, direction of differentiation/derivation; blue arrows, counterpart relationships; question mark, a speculation to be further verified. Source data are provided as a Source Data file. See also Supplementary Fig. 8.

In both naive and primed hESCs, we found that WNT activation was required for ExM specification in the presence of BMP4, and that inhibition of WNT signaling led to commitment to the AM fate with near-complete depletion of ExMs (Fig. 7l). When BMP4 and CHIR were present together, both naive and primed hESCs exhibited bidirectional differentiation potential into ExM and AM, with naive hESCs predominantly differentiating into ExM but primed hESCs largely adopting the AM fate (Figs. 3j, 5o). On this basis, exogenous activation of the Nodal signaling promoted efficient ExM specification in both naive and primed hESCs by abolishing AM differentiation (Fig. 7l). In short, WNT and Nodal signaling play crucial roles in the specification of ExM from hESCs in the context of BMP activation; that is, ExM specification is WNT-dependent, and Nodal signaling promotes ExM differentiation by blocking the AM fate (Fig. 7l). Findings in previous researches have revealed that BMP signaling induces both WNT-dependent mesodermal lineage commitment and WNT-independent AM fate in hESCs[25]; WNT signaling is necessary and sufficient to induce PS fate[23–25]; and Nodal signaling also promotes PS formation[23,29] and inhibits AM specification[30]. Given that ExM specification undergoes a PSLI, it is logical that dual activation of WNT and Nodal signaling results in the efficient differentiation of hESCs into ExMs in the context of BMP activation.

## Methods

### Ethics statement

The research about human embryo is a continuation of our previous work and had been approved by the Medicine Ethics Committee of The First People's Hospital of Yunnan Province (KHLL2020-KY064)[3,26]. All donated embryos were supernumerary frozen embryos after IVF clinic treatment. The informed consent process for embryo donation complied with International Society for Stem Cell Research (ISSCR) Guidelines (2021) and Ethical Guidelines for Human Embryonic Stem Cell research (2003) jointly issued by Ministry of Science and Technology and Ministry of Health of People's Republic of China. All the donor couples had signed the informed consent for voluntary donations of supernumerary embryos for this study. No financial inducements were offered for the donations.

All differentiation experiments of human embryonic stem cells (hESCs) in this study complied with the 2021 ISSCR Guidelines[41]. Work with hESCs to model early human development was approved by the Medical Ethics Committee of Yunnan Key Laboratory of Primate Biomedical Research (LPBR-YX003) and also by the Medical Ethics Committee of Kunming University of Science and Technology (KMUST-MEC-206 and KMUST-MEC-2023-034).

### Embryo manipulation

To control the variability of the blastocysts, we evaluated the quality of the embryos before use. According to the Gardner's scoring system[42], thawed blastocysts were given numerical scores from 1 to 6 based on their expansion degree and hatching status. The blastocysts with expansion and hatching status above 3 and with visible

inner cell mass above grade B were used in the study. Frozen-thawed human blastocysts (5-6 days post-fertilization) were gently treated by acidic Tyrode's solution (Sigma, T1788) to remove the zona pellucida[26]. In vitro 3D culture and frozen section staining of human embryos were performed according to previously published methods[26].

### hESC culture

Naive and primed hESCs were used in this study. Unless otherwise specified, all hESC culture experiments were performed in a humidified incubator under 21% O₂ and 5% CO₂ at 37 °C. hESC lines were routinely checked for mycoplasma contaminations using MycoAlert Mycoplasma Detection Kit (LONZA, LT07-318) every two weeks, and all cell lines used in this study have been ruled out of mycoplasma contamination. hESC lines were authenticated by genomic PCR, immunostaining, bulk RNA-seq, and in vitro differentiation tests.

### Naive hESCs

The three naive hESC lines (AIC-N1, AIC-N2, and AIC-N4) used in this study were established in our previous study[3] and propagated on mouse embryonic fibroblast (Millipore, PMEF-CFL) feeder cells in AIC-N[3] or PXGL[20] medium. AIC-N medium was composed of modified N2B27 medium supplemented with 10 ng/ml Activin-A (Peprotech, 120-14E), 2 μM IWP2 (Selleck, S7085), 0.3 μM CHIR99021 (Selleck, S2924), 1 μM PD0325901 (Selleck, S1036), 2 μM Gö6983 (TOCRIS, 2285), and 10 ng/ml recombinant human LIF (Peprotech, 300-05). PXGL medium was composed of modified N2B27 medium supplemented with 1 μM PD0325901, 2 μM XAV939 (Sigma, X3004), 2 μM Gö6983, and 10 ng/ml recombinant human LIF. 500 ml modified N2B27 medium was composed of 240 ml DMEM/F12 (Thermo Fisher Scientific, 10565-018), 240 ml Neurobasal (Thermo Fisher Scientific, 21103-049), 5 ml N2 supplement (Thermo Fisher Scientific, 17502-048), 10 ml B27 supplement (Thermo Fisher Scientific, 17504-044), 0.5% GlutaMAX (Thermo Fisher Scientific, 35050-061), 1% nonessential amino acids (NEAA, Thermo Fisher Scientific, 11140-050), 0.1 mM β-mercaptoethanol (Sigma, M7522), 0.38 ml 7.5% BSA (Sigma, A1933), 50 μg/ml L-ascorbic acid 2-phosphate (Sigma, A8960), 0.5 ml Chemically Defined Lipid Concentrate (Thermo Fisher Scientific, 11905-031), 12.5 μg/ml Insulin (Roche, 11376497001), 1.25 ml Sodium Pyruvate (Thermo Fisher Scientific, 11360-070), 0.02 μg/ml Progesterone (Sigma, P8783). AIC-N or PXGL medium was supplemented with 10 μM Y27632 (Selleck, S1049) during the maintenance of naive hESCs. Naive hESCs were passaged by single cell dissociation with 50% TrypLE (Thermo Fisher Scientific, 12605-028) every 4-5 days at a 1:5 to 1:10 split ratio. Naive hESCs grown in PXGL medium were cultured under 5% O₂. Unless otherwise specified, naive hESCs were typically cultured in AIC-N medium under 21% O₂.

### Primed hESCs

The primed hESC lines used in this study include hES1, h1, and H9 hESC lines[28], as well as the MESP1-mTomato knock-in reporter H9 hESC line[31].

Primed hESCs were cultured on Matrigel (Corning, 354277)-coated dishes/plates in the AIC[28] or TeSR-E8 (E8; STEMCELL Technologies, 05990) or mTeSR1 (STEMCELL Technologies, 85850) medium. Briefly, the AIC or E8 or mTeSR1 medium was changed every two days and primed hESCs were passaged every 3-4 days at a 1:10-1:20 split ratio by single-cell dissociation with 50% TrypLE. AIC or E8 or mTeSR1 medium was supplemented with 5 μM Y27632 on the first day after passaging. AIC medium was composed of modified N2B27 medium supplemented with 10 ng/ml Activin-A, 2 μM IWP2, and 0.6 μM CHIR99021. Unless otherwise specified, primed hESCs were typically cultured in AIC medium.

### Adherent differentiation of hESCs

Naive or primed hESCs were digested into single cells with 50% TrypLE and inoculated in the following induction media at a density of 1-1.5×10^4 cells/cm^2 onto Matrigel-coated dishes or plates, without mouse embryonic fibroblast feeder cells. The media were refreshed every two days until the end of the assay. CB (CHIR + BMP4) induction medium was composed of modified N2B27 medium supplemented with 50 ng/mL recombinant human FGF4 (Peprotech, 100-31-25), 1 μg/mL heparin sodium salt (Sigma, H3149), 10 μM Y27632, 2 μM CHIR99021, and 10 ng/mL BMP4 (R&D Systems, 314-BP-050). CBA (CHIR + BMP4+Activin-A) induction medium was composed of CB induction medium supplemented with Activin-A (5 ng/mL). IB (IWP2 + BMP4) induction medium was formulated by substituting IWP2 (3 μM) for CHIR99021 in the CB medium. IBA (IWP2 + BMP4+Activin-A) induction medium was composed of IB induction medium supplemented with Activin-A (5 ng/mL). Routinely, cells are induced under CB or IB conditions for 4 days and under CBA conditions for 5 days.

### Extended culture of CB/CBA-treated hESCs

For CB-treated primed hESCs (CBPs), predominantly composed of amnion (AM)-like cells, we did not perform further extended culture.

For CB-treated naive hESCs (CBNs), predominantly composed of extraembryonic mesoderm-like cells (ExMs), we employed flow cytometry to isolate E-Cadherin⁻ cells and subsequently plated them in human trophoblast stem cell medium (hTS-M)[43] for further extended culture.

For CBA-treated primed (CBAPs) and naive (CBANs) hESCs, which were almost devoid of AM-like cells, we digested them into single cells on day 5 and plated them in hTS-M for further extended culture.

Generally, the CBNs, CBANs, and CBAPs grown in hTS-M were passaged every 3-5 days at a 1:3 to 1:5 split ratio by single-cell dissociation with 50% TrypLE. hTS-M was composed of the following ingredients: DMEM/F12 supplemented with 0.1 mM 2-mercaptoethanol, 0.2% FBS (BI, 04-002-1 A), 0.3% BSA, 1% ITS-X supplement (Thermo Fisher Scientific, 51500-056), 50 μg/mL L-ascorbic acid 2-phosphate, 50 ng/mL EGF (R&D Systems, 236-EG-01M), 2 μM CHIR99021, 0.5 μM A83-01 (Tocris, 2939), 1 μM SB431542 (Cellagen, C7243), 0.8 mM Valproic acid (VPA, Sigma, P4543), and 5 μM Y27632.

### Derivation of AAVS1-Knockin hESC lines

2.5×10^5 hESCs (primed hESCs or naive hESCs) were electroporated with 1 μg of donor plasmid AAVS1-CAG-hrGFP (Addgene, #52344) or AAVS1-Pur-CAG-mCherry (Addgene, #80946), and 1 μg of sgRNA-CAS9 expression vector by 4D-Nucleofector (Lonza) using P3 Primary Cell 4D-Nucleofector X kit (Lonza). Transfected cells were plated onto irradiated DR4 MEFs (the Cell Bank of the Chinese Academy of Sciences, Shanghai, China; https://www.cellbank.org.cn) in AIC (for primed hESCs) or AIC-N (for naive hESCs) media. Three days later, cells were selected and expanded in AIC or AIC-N media with puromycin (0.25 μg/ml) for 2 weeks, hrGFP/mCherry-positive and puromycin-resistant clones were picked and identified by PCR.

### Generation of TCF/LEF:H2B-GFP cell lines

To generate TCF/LEF reporter-hESC lines, we first constructed a lentiviral plasmid vector with 8XTCF/LEF-minP-H2B-eGFP (pLV.TCF-LEF RE-H2B-EGFP.PGK.Puro). The gene sequence of 8XTCF/LEF-minP (GenBank: JX099537.1)-H2B-EGFP was synthesized and ligated into a lentiviral plasmid with a constitutively active puromycin selection from PackGene Biotech (XL02C, https://www.packgene.cn/products/plasmids-product/). We co-transfected HEK293T cells with pMD2.G, psPAX2, and pLV.TCF-LEF RE-H2B-EGFP.PGK.Puro plasmid to produce lentivirus. Subsequently, AIC-N2 hESCs cultured in AIC-N medium were transduced with the lentivirus. After 48 hours, the infected AIC-N2 hESCs were selected and expanded in AIC-N medium containing puromycin (0.25 μg/ml) for 2 weeks. Three clonal lines were manually picked and further expanded. H2B-EGFP expression was confirmed by fluorescence microscopy following treatment with CB induction medium.

### Time-lapse live imaging of MESP1-reporter cell lines

Primed MESP1-mTomato knock-in reporter hESCs were inoculated at a density of 2×10^5 cells/well onto Matrigel-coated 24-well plate in CBA induction medium. 24 hours after cell seeding, the medium was refreshed wtih 2 ml/well fresh medium and the 24-well plate was transferred to the humidified chamber of CellVoyager CQ1 Benchtop (Yokogawa) to monitor the expression dynamics of MESP1. Confocal time-lapse live imaging was performed using a 10× objective lens with a appropriate laser/filter for mTomato (561 nm). The cells were imaged at 5-hour intervals for 4 days under 21% O₂ and 5% CO₂ at 37 °C.

### Flow cytometry

To detect specific markers of ExM or AM lineages using flow cytometry, naive and primed hESC derivatives grown under different induced conditions were dissociated into single cells with 50% TrypLE, centrifuged, and washed with ice-cold DPBS containing 1% FBS (BI, 04-001-1 A). Dissociated single cells were fixed with 4% paraformaldehyde at room temperature for 10 minutes, then washed three times with ice-cold DPBS containing 1% FBS and 100 mM glycine. Subsequently, permeabilization and blocking were performed with DPBS containing 0.2% Triton X-100 and 3% BSA at room temperature for 15 minutes. The live-cell staining for E-cadherin was performed without fixation and permeabilization. Cells were incubated at 4 °C for 30 minutes with conjugated primary antibodies and their isotype control diluted in DPBS containing 1% BSA, and then washed three times with ice-cold DPBS containing 1% FBS. For unconjugated primary antibodies and their isotype control, cells were further stained at 4 °C for 30 minutes with secondary antibodies diluted in DPBS containing 1% BSA, followed by three washes with ice-cold DPBS containing 1% FBS. For the sorting of MESP1-mTomato-positive primitive streak-like intermediates, wild-type CBAPs/CBANs differentiated for the same duration were used as negative controls. Flow cytometry was carried out using a FACSAria III, and the data were analyzed using FlowJo software. The antibodies were listed in Supplementary Table 1.

### Immunofluorescence staining

All adherently growing cells in the study were fixed with 4% paraformaldehyde for 20 minutes at room temperature and washed three times with DPBS containing 100 mM glycine. For 3D-cultured human embryos, the frozen section staining was performed according to previously published methods[26]. Briefly, embryos were transferred and fixed in 4% paraformaldehyde at 4 °C for 3 hours, washed three times with DPBS containing 100 mM glycine, dehydrated overnight in DPBS containing 20% sucrose at 4 °C, embedded in O.C.T. (Sakura Finetek, 4583), and then sectioned using a Leica frozen slicer at a thickness of 10 μm. After permeabilization and blocking with DPBS containing 0.2% Triton X-100, 100 mM glycine, and 3% BSA at room temperature for 60 minutes, the cells or sections were incubated with primary

antibodies at 4 °C overnight, washed thrice with DPBS containing 0.05% Tween-20, incubated with secondary antibodies for 2 h at room temperature and washed thrice with DPBS containing 0.05% Tween-20. DAPI (Sigma, 32670) was used for staining the nuclei. The antibodies are listed in Supplementary Table 1. Images were captured using a Leica SP8 laser confocal microscope or a Nikon AX laser confocal microscope.

The fluorescence intensity was analyzed using Fiji software, and the mean fluorescence intensities of individual cells were taken as the total fluorescence intensity within the cell region divided by the area of that cell. The fluorescence intensity profile was generated using the Line Plot function in FIJI software: (1) draw a line using the Line tool; (2) run the command Analyze > Plot Profile to generate an intensity line plot; and (3) click List to display and export the plot values. The mean fluorescence intensities were measured using FIJI software: (1) select the region and adjust the threshold through "Image-Adjust-Threshold"; (2) set the parameters through "Analyze-Set Measurements"; and (3) run "analytic-measure" and export the plot values. To quantify the proportions of different cell types via immunofluorescence, at least three fields were randomly captured using a confocal microscope with a 10× objective lens for each experiment. Total cell numbers (DAPI-stained nuclei) and the numbers of different cell types (expressing specific markers) were quantified using the spots tool in Imaris software (version 10.0, Oxford Instruments). The graphs were plotted using GraphPad Prism 9.0.

## Bulk RNA sequencing and data analysis

The following cell types were collected for bulk RNA sequencing: (1) naive (AIC-N1, AIC-N2, and AIC-N4) and primed (Primed H9, Primed hES1, and Primed h1) hESC derivatives grown under CB, CBA, and IB conditions, including CBNs from day 1 to day 4, CBPs from day 1 to day 4, CBANs from day 1 to day 5, CBAPs from day 1 to day 5, and D4 IBNs; and (2) expandable ExMs grown in hTS-M, including CBNs, CBAPs, and CBANs at P5/P10.

Adherently growing cells on Matrigel were collected by dissociating into single cells using 50% TrypLE and were washed twice with DPBS. Total RNA was isolated with the TRIzol™ Reagent (Thermo Fisher Scientific, 15596018). Library construction and sequencing were performed by Annoroad Gene Technology (http://www.annoroad.com/). Libraries were sequenced with Illumina NovaSeq 6000 and 150 bp paired-end reads were generated. Publicly available datasets from naive (AIC-N)[3] and primed (AIC)[28] hESCs were included in this study. Reads were aligned to human genome (UCSC GRCh38) using HISAT2 (v2.2.1)[44]. The counts and FPKM values for each gene were calculated with StringTie (v2.1.1)[45]. Principal components analysis was performed using prcomp function from the R stats package based on the gene matrix with FPKM ≥ 1 in at least one sample. Heatmaps were generated using pheatmap package from the R software.

Correlation analysis was performed by Spearman correlation based on the gene matrix with FPKM ≥ 1 in at least one sample. Differentially expressed genes were detected by the package DESeq2 (v1.42.1)[46] in the R software. An adjusted $p$-value < 0.05 and an absolute value of the $\log_2$ Fold Change ≥ 1 and FPKM ≥ 1 in at least one sample were used as the threshold for declaring gene expression differences as being significant. Gene Ontology (GO) and Kyoto Encyclopedia of Genes and Genomes (KEGG) analyses were conducted with the KOBAS.

## Single cell dissociation, RNA sequencing and data processing

The following cell types were collected for single-cell RNA sequencing (scRNA-seq) using the 10X Genomics platform: mCherry-labeled male naive (AIC-N4) hESCs and hrGFP-labeled female primed (AIC-H9) hESCs, as well as their derivatives grown under CB and CBA conditions, including CBNs from day 1 to day 4, CBPs from day 1 to day 4, CBANs on day 5, and CBAPs on day 5.

For mCherry-labeled male AIC-N4 hESCs grown on feeders, the colonies were detached from the feeders by exposure to Collagenase type IV for 60 to 90 minutes. The detached colonies were then collected, transferred into 50% TrypLE, incubated for 7 minutes at 37 °C, and gently dissociated into single cells by pipetting up and down. Adherently growing hrGFP-labeled female primed H9 hESCs on Matrigel and all hESC derivatives grown under CB and CBN conditions were collected by dissociating into single cells with 50% TrypLE. All dissociated single cells were filtered through a 20 μm cell strainer, centrifuged, suspended in DPBS containing 0.04% BSA and counted using a Countstar automatic cell counter. To minimize batch effects and costs in scRNA-seq, we collected the following sample pairs as pooled samples by mixing equal cell numbers: mCherry-labeled male AIC-N4 hESCs and hrGFP-labeled female primed (AIC) H9 hESCs (AIC-N4/Primed H9), CBN4-D1 and CBP9-D1 (CB1), CBN4-D2 and CBP9-D2 (CB2), CBN4-D3 and CBP9-D3 (CB3), CBN4-D4 and CBP9-D4 (CB4), and CBAN4-D5 and CBAP9-D5 (CBA5).

Single cell suspension of all pooled samples were loaded into the 10x Genomics Chromium system within 30 minutes after dissociation. Chromium Single Cell 3′ v3.1 libraries were prepared according to the manufacturer's instructions. Libraries were sequenced with a minimum coverage of 30, 000 raw reads per cell on an Illumina NovaSeq 6000 with 150-bp paired-end sequencing, which was performed by Annoroad Gene Technology (http://www.annoroad.com/). The sequences of *hrGFP* and *mCherry-WPRE* (according to the donor plasmid) were added to the Homo sapiens genome GRCh38 according to the cellranger mkref pipeline. Sequencing data was aligned and quantified using the Cell Ranger Pipeline v7.0.1 (10x Genomics) against the GRCh38 reference genome with *hrGFP* and *mCherry-WPRE*. scRNA-seq data were filtered based on number of expressed genes and expression level of mitochondrial genes (below 15%). Cell doublets were removed by DoubletFinder (v2.0.3)[47] with assuming multiplet rate according to the loaded cells number (refer to Multiplet Rate Table provided in the 10x Genomics User Guide). Gene expression levels (normalized and natural-log (log1p) transformed value) of *RPS4Y1*, *hrGFP*, and *mCherry-WPRE* were used to determine the source of different cell types in pooled samples. Cells positive for *mCherry-WPRE* (value > 0) and negative for *hrGFP* (value = 0) were defined as derivatives from the mCherry-labeled male AIC-N4 hESCs and further filtered with *RPS4Y1* expression (value > 1), and cells positive for *hrGFP* (value > 0) and negative for *mCherry-WPRE* (value = 0) were defined as derivatives from hrGFP-labeled female primed H9 hESCs and further filtered with *RPS4Y1* expression (value < 1). Double positive or double negative cells were removed for further analyses.

Further analyses were performed using Seurat package (v4.3.0.1)[48]. The raw counts were normalized and scaled with default parameters. Top 2000 most variable genes were identified and used for dimensionality reduction with PCA and followed with non-linear dimensionality reduction using UMAP. The scRNA-seq data from differentiation time courses for CBNs or CBPs were combined with merge() function. Cell types were defined based on the lineage markers and clusters identified through FindClusters() function. Data was visualized with the UMAP dimensionality reduction. Differentially expressed genes were identified with the FindAllMarkers() function in Seurat and filtered with $P$adj of Wilcoxon's rank-sum test <0.05, $\log_2$ (FC) > 0.25 and expressed in > 25% of cells of the given cluster. GO and KEGG analyses were conducted with the KOBAS. Single-cell pseudotime trajectories were constructed with Monocle 2 (v2.18.0)[49] and Slingshot (v1.8.0)[50]. RNA velocity analysis was performed by scVelo (v0.3.1)[51]. Co-expression network analysis was performed by hdWGCNA (0.4.00)[36].

To compare our datasets with the publicly available datasets, IntegrateData() function was used to integrate the scRNA-seq data from this study with published scRNA-seq data, including that from

human CS7 gastrulating embryo[4], cultured human peri-implantation embryos[3], and naive hESC derivatives in ASECRiAV[12].

## Statistics and reproducibility

Statistical tests were performed on GraphPad Prism 9 software and Microsoft office Excel 2019. Data were checked for normal distribution and equal variances before each parametric statistical test was performed. Where appropriate, t-tests were performed with Welch's correction if variance between groups was not equal. ANOVA tests were performed with a Dunnett's multiple comparisons test if variance between groups was not equal. Error bars represent standard deviation in all cases, unless otherwise noted. Figure legends indicate the number of independent experiments and statistical subjects performed in each analysis.

Unless otherwise noted (MESP1-mTomato knock-in reporter hESC line and AIC-N2 TCF/LEF:H2B-GFP reporter hESC line, for which only one cell line was available), all representative bright-field and staining results in this study were reproducibly obtained in at least three independent hESC lines. Reporter cell line data were consistently replicated in ≥ 3 technical repeats.

## Reporting summary

Further information on research design is available in the Nature Portfolio Reporting Summary linked to this article.

## Data availability

The raw sequence data from our study have been deposited in the Genome Sequence Archive in National Genomics Data Center under the BioProject accession code PRJCA037022. This paper does not report original code, publicly available tools were used in data analysis as described. Any additional information required to reanalyze the data reported in this paper is available from the lead contact upon request. Previously published datasets that were used in this study: bulk RNA-seq datasets from primed hESCs (GSE145163); bulk RNA-seq datasets from naive hESCs and scRNA-seq datasets from cultured human embryos (PRJCA017779); scRNA-seq datasets from human CS7 gastrulating embryo (GSE193007); and scRNA-seq datasets from naive hESC derivatives in hTS-M ASECRiAV (GSE204779). Source data are provided with this paper.

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

## Acknowledgements

This work was supported by the National Key Research and Development Program of China (2022YFA1103100 to T.L., 2020YFA0112700 to Z.A., and 2021YFA0805700 to T.L.), the National Natural Science Foundation of China (32130034 to T.L., 32360177 to Z.A., 32260176 to B.N., 32360178 to Y.Y., and 32470859 to Z.A.). We are grateful to Dr. Jie Na and her colleagues for providing the MESP1-mTomato knock-in reporter hESCs; Dr. Chenyang Si and Dr. Yu Kang for their assistance with live-cell imaging; and Dr. Chengxin Wu and Ms. Min Yan from Advanced Imaging Platform of Institute of Primate Translational Medicine, Kunming University of Science and Technology for their outstanding support in confocal fluorescence imaging.

## Author contributions

Z.A., T.L. and W.J. conceptualized this project and supervised the overall experiments. Z.A., T.L., B.N. and Z.C. designed the experiments and wrote the manuscript. Z.A., B.N., Y.H., Y.W., C.Z., X.W., R.K. and H.C. established the differentiation protocol of extraembryonic mesoderm-like cells, performed the experiments, and analyzed the data. G.S. and L.X. performed human embryo culture. D.W. and Y.Y. analyzed sequencing data. All authors discussed and commented on the manuscript.

## Competing interests

The authors declare no competing interests.
