## [Transparent Peer Review file · Nature Communications]

Deciphering signaling mechanisms and developmental dynamics in extraembryonic mesoderm specification from hESCs

Corresponding Author: Dr Zongyong Ai

Version 0:

Reviewer comments:

Reviewer #1

(Remarks to the Author)

Liu et al. describe the generation of extra-embryonic mesoderm from human pluripotent stem cells, highlighting the signaling cues and transcriptional pathways involved. Importantly, the authors use robust molecular criteria to define extra-embryonic mesoderm and demonstrate how these criteria overlap with reference embryo data. They also show that the generation of extra-embryonic mesoderm follows a pathway with intermediates consistent with the primitive streak cells. I find the data and analysis to be very well executed and compelling. Furthermore, shedding light on the generation and specification of human extra-embryonic mesoderm is a highly relevant area of research.

The only potential limitation of the study is the absence of a functional assay for the generated extra-embryonic mesoderm cells. However, as it stands, the work is adequate for publication in Nature Communications.

I also suggest the authors to leave open the possibility of additional routes for the generation of extra-embryonic mesoderm.

Reviewer #2

(Remarks to the Author)

Niu et al.

Niu et al. describe the differentiation and propagation of extraembryonic mesoderm (ExM) cells from human naïve and primed pluripotent stem cultures in two dimensions. The authors use immunofluorescence and scRNA-seq to follow the differentiation process and compare the end cells and the intermediates to cells from human embryos. They define the roles of BMP WNT signaling in ExM specification. A key conclusion is that cells go through a primitive streak-like intermediate on the pathway to ExM. The authors carry out a detailed comparison of ExM derived from naïve versus primed cells.

The extraembryonic mesoderm is a critical component of many supporting tissues during development and its origins in primates are not well understood. One previous study demonstrated differentiation of human ExM from naïve pluripotent stem cells (Pham et al, cited by the authors) ; a recent work showed the dependence of this pathway on WNT signaling (<https://doi.org/10.1111/cpr.13761>) as did the authors' previous study (Ai et al), and ExM formation has been described in numerous post-implantation human embryo models though the origins of the tissue have not been deeply studied in this context. Novel aspects of this work include a rapid and efficient induction of ExM, the extensive comparison of ExM derived from naïve and primed cells and the identification of a primitive streak like intermediate during ExM differentiation. Further differentiation of these cells or functional characterization (for example extracellular matrix formation) are not studied.

The findings regarding the presence of putative ExM precursor cells with a primitive streak phenotype are interesting, but the inferences regarding the primitive streak as an intermediate to ExM are potentially confusing and do not have a very strong experimental basis. The primitive streak is not a cellular state but a structure that emerges at the onset of gastrulation, and it is widely agreed that ExM is identifiable before this point in primates, namely around CS5. Indeed, this is a feature that distinguishes primate from mouse development. The authors study induction of ExM in 2D culture which as discussed below may not reflect a normal developmental sequence. If the authors are contending that a primitive streak structure appears

prior to gastrulation to give rise to ExM, they need to provide more convincing microscopic images than those shown in Figure 4 and S4. A more likely interpretation of the authors' data is that the ExM intermediate recapitulates a gene expression program also used in formation of the primitive streak. However, it should also be noted that scRNA-seq data do not prove lineage relationships; they are suggestive only. Clonal isolation in vitro of PS-like cells (using a cell surface antibody or reporter cell line) followed by a demonstration of their conversion to ExM could address this point more directly. See below for further considerations around this issue.

Specific points

1. L154-has EnExM any counterpart in the normal embryo? Could its phenotype reflect hypoblast origin of ExM as has been hypothesized previously?
2. L168- in Reference 3 (Ai et al.), the authors clearly identified two populations of ExM-one that emerges early, at E11, before cells of the primitive streak which appear at E13-14, and one that appears later, contemporaneously with the PS (data from Fig. S1 Ai et al.). Is it possible that the ExM produced in the 2D cultures represents a later wave of ExM that actually originates from the streak, and that the earlier population is not represented in the 2D model?
3. L270 emergence of PS like cells on D1 of the protocol indicates that the 2D system is not reflective of any normal developmental sequence; naïve epiblast does not directly convert to PS.
4. L283 what is the relationship if any between ExM1 and ExM2 described in this study and the populations with the same names in the embryo study in Ai et al. (point 1 above).
5. L311-this is a confusing statement, see FigS1 of Ai et al. RNA velocity does not directly establish lineage relationships.
6. L317-as noted above, the 2D system, though an efficient means of differentiating ExM, does not reflect a normal developmental sequence.
7. L329-342-the conclusion that ExM arises from PS in the embryo at E11 (PS was present at very low levels in the authors' previous study Ai et al. at this time point) is based on 2 embryos and a handful of T+ cells (scattered around epiblast) a subset of which co-express ExM markers. These data may be consistent with the presence of cells precociously expressing primitive streak markers at E11 but do not convincingly illustrate the presence of a primitive streak. It would be informative to see and E13 embryo similarly stained for comparison.
8. L338-again the statement that no ExM is present at E11, but that PS is, contradicts much prior data including that in Ai et al.
9. L448-548-the differences between the trajectories of naïve v primed cells are clearly documented, but the developmental significance of these findings is unclear.
10. It is possible given the data presented here that a small transient population of cells with a primitive streak like phenotype give rise to ExM. This is feasible; it is certainly conceivable that the molecular program for embryonic mesoderm and ExM induction is similar. It is also possible that a subpopulation of ExM derives from PS after around E13. Perhaps the data are consistent with either interpretation, but they do not prove these possibilities directly. I do not think the authors can assert that all ExM derives from the primitive streak, if that is what they are meaning to claim.

Reviewer #3

(Remarks to the Author)

In this review, Niu et al develop a new protocol to generate extra-embryonic mesoderm (ExM) cells from human ESCs and identify the signals that drive ExM fate from different pluripotent states. The biology of human ExM is poorly understood. Only very recent publications have begun shedding light on the mechanisms that control ExM specification. Therefore, this article will be of interest to the community. I support the publication of the article, but several issues need to be addressed.

Main comments:

- The authors use AIC-N to maintain human ESCs in a naïve pluripotent state, and AIC to preserve primed pluripotency. While these culture conditions have already been characterized by the authors, they are not broadly used by the community. Therefore, it would be important to first briefly explain them in the text (only the original references are provided), and to validate the key findings of the manuscript using alternative naïve and primed conditions. This would strengthen the relevance of the findings.
- Figure 1: The authors use CN to induce ExM from human ESCs, and then human TSC medium for long-term culture. It is not clear whether CN can sustain the expansion of ExM. If yes, do the ExM cells retain their earlier ExM identity in this condition?
- Figure 2: The finding that Wnt signalling needs to be inhibited to make trophoblast is surprising, as Wnt activation is needed to preserve TSC identity. How is this explained? Are the trophoblast cells eventually differentiating to an extra-villous trophoblast identity when IWP2 is used (prolonged IBN culture)? Do the cells acquire a bona fide TSC identity if instead they are switched to human TSC medium after 4 days in IBN?
- Figure 4: the human embryo data is weak. The images provided are saturated. It is difficult to understand which structures are present, and the embryos seem to be disorganized. Some of the nuclei that are highlighted in Figure S4D seem to be DAPI fragments/debris, and all the arrowheads obstruct the view. Moreover, data from human embryos at day 13 showing ExM cells is needed to fully characterize ExM specification in in vitro cultured embryos. Finally, it is not clear how the authors put their findings in the context of embryo development. In the discussion, they mention "precocious gastrulating cells". What does this mean? Either there is gastrulation or there isn't. Do the authors see the hallmarks of gastrulation already on day 11? Is there basement membrane breakdown? Is there delamination and loss of epithelial integrity of epiblast cells? Does this happen in a polarised manner within the epiblast? What is the meaning of the Brachyury+ cells that are detected on day 11?
- Figure 5: in the CBP condition the authors mention that cells undergo an amnion to a trophoblast fate transition. Similarly to my previous comments, are bona fide trophoblast cells being specified? If switched to a human TSC medium, can they be

maintained long-term? Generally, throughout the text, amnion and trophoblast are grouped together as one fate, alternative to the ExM. Why are they grouped? Is this because both cell types are specified? Or is it because technically the authors have not been able to distinguish whether the cells are amnion or trophoblast, or even a mixed identity? The finding that primed cells require Activin-A for ExM specification while naïve cells don't is surprising, as it would be likely that naïve cells transition through a primed state before making ExM (potentially post-EPI state in the UMAP plot (Figure 5J)). Is this the case? It would be informative to compare the dynamics of ExM induction from naïve and primed to better understand the findings. Does the addition of Activin-A block the effect of IWP2?

- Figure 7: the description of the findings should be more conceptual. The authors provide lists of genes and lists of GO terms that globally lack meaningful information. What is the conclusion that is drawn from all the GO terms and differentially expressed genes? This also applies to the description of other figures but is especially relevant for Figure 7.
- Methodology: several immunofluorescence images appear saturated, and it is difficult to see any clear pattern of localization (for example active beta-catenin in panel 2C). Moreover, the quantifications provided are not appropriate (i.e., panel 2B). Fluorescence intensity levels need to be measured in single nuclei using unsaturated images and the data needs to be statistically analysed. This would also serve as a validation of the experimental tools. For example, is there a positive correlation between the nuclear fluorescence intensity of beta-catenin and H2B-GFP (from the TCF/LEF:H2B-GFP line)?

Minor comments:

- Line 143-144: this sentence is incorrect. ExM cells have never been derived from a human embryo.
- Where do PGCs come from based on the pseudotime analysis?
- Some panels are not well organized, are difficult to interpret, or are not properly explained (either in the text or the legends). For example, in Figure 2A it is not clear if the magnifications are done in any specific area of the non-magnified images. The non-magnified images are not very informative. Figure S3c mentions hrGFP-labeled female ESCs, what are these cells? What are they used for? Similarly, why are the authors using mCherry-labelled male cells for the experiments? Some UMAPs are missing axes. There is no consistency in the nomenclature. For example, in Figure 5B the authors refer to "primed cells" but in Figure 5C they refer to the same cells as "AIC".
- Why one more day was used for the protocols that entailed the addition of Activin-A?

Version 1:

Reviewer comments:

Reviewer #2

(Remarks to the Author)

In this revised manuscript, the authors have provided some additional experimental findings, in particular direct fate analysis MESP1+ cells, and new analyses of scRNA-seq data, and have refined some of the interpretations of their data so as improve the precision of the conclusions of the study. While questions remain over the the origin and function of the ExM populations, the authors' work will inform future studies in this rapidly evolving field.

Reviewer #3

(Remarks to the Author)

The authors have made a significant effort to address the reviewers' comments. Most of the comments are successfully addressed, but the relevance of the findings for embryo development is still questionable. As it stands, the human embryo data is very weak, and no conclusions regarding the physiological relevance of the findings can be drawn. The authors have not provided additional experiments, as requested both by reviewer 2 and me. Either the authors perform additional experiments in human embryos to fully characterize what is happening in the embryo in terms of primitive streak and ExM differentiation, or they remove the embryo data and their claims of physiological relevance. In addition, there are a couple of points that need further clarification:

1. Response of other naïve and primed human ESCs to CB medium: the authors have addressed my initial comment. However, Figure 5n is missing naïve ESCs (AIC and PXGL) as a control. This is needed to conclude that E8-cultured primed ESCs behave more similarly to mTESR/AIC primed human ESCs than to naïve ESCs.

2. Methods: the authors mention that they have "reduced the fluorescence intensity of the immunofluorescence images". How did they achieve that? A saturated image cannot be converted into an unsaturated image. It seems they did not repeat the experiments, which would be needed if the raw images were saturated as quantifications would not be possible. The methods section should also include details on how the quantification of the immunofluorescence images was done.

Dear Reviewers,

On behalf of my co-authors, we appreciate you very much for the constructive comments and suggestions on our manuscript entitled “**Deciphering signaling mechanisms and developmental dynamics in extraembryonic mesoderm specification from hESCs**”. Your comments and suggestions are highly insightful and enable us to greatly improve the quality of our manuscript.

In the revised manuscript, we provided more data to address the main concerns raised by you, and made revisions to the previous manuscript based on your suggestions.

The “**Point-by-point response to the reviewers’ comments**” was presented in the following:

Rebuttal: Manuscript NCOMMS-24-63802

We would like to sincerely thank the reviewers for their constructive comments and suggestions, which we have used as the basis for revising our manuscript.

Comments from reviewer(s): Reviewer #1 (Remarks to the Author):

Niu et al. describe the generation of extra-embryonic mesoderm from human pluripotent stem cells, highlighting the signaling cues and transcriptional pathways involved. Importantly, the authors use robust molecular criteria to define extra-embryonic mesoderm and demonstrate how these criteria overlap with reference embryo data. They also show that the generation of extra-embryonic mesoderm follows a pathway with intermediates consistent with the primitive streak cells. I find the data and analysis to be very well executed and compelling. Furthermore, shedding light on the generation and specification of human extra-embryonic mesoderm is a highly relevant area of research.

The only potential limitation of the study is the absence of a functional assay for the generated extra-embryonic mesoderm cells. However, as it stands, the work is adequate for publication in Nature Communications.

I also suggest the authors to leave open the possibility of additional routes for the generation of extra-embryonic mesoderm.

Response

We sincerely appreciate your encouraging comments and constructive suggestion.

Given the study's primary focus on the developmental dynamics and signaling mechanisms of extraembryonic mesoderm cells (ExMs), as well as the influence of the initial pluripotent state on ExM specification, the functional exploration of ExMs is beyond the scope of this research. In future work, we will investigate the function of ExMs from both naive and primed hESCs.

Indeed, the routes for ExM generation remain elusive, and we strongly agree with your suggestion. Beyond the pluripotency-to-PSLI-to-ExM trajectory elucidated in this study, there may be other potential ExM generation routes, such as the hypoblast route. In this study, we conclude that under CB or CBA conditions, hESCs mainly undergo PS-like intermediates (PSLIs) to differentiate into ExMs, and we do not rule out the possibility of a small number of ExMs being generated through

alternative routes.

Comments from reviewer(s): Reviewer #2 (Remarks to the Author):

Niu et al. describe the differentiation and propagation of extraembryonic mesoderm (ExM) cells from human naive and primed pluripotent stem cultures in two dimensions. The authors use immunofluorescence and scRNA-seq to follow the differentiation process and compare the end cells and the intermediates to cells from human embryos. They define the roles of BMP and WNT signaling in ExM specification. A key conclusion is that cells go through a primitive streak-like intermediate on the pathway to ExM. The authors carry out a detailed comparison of ExM derived from naive versus primed cells.

The extraembryonic mesoderm is a critical component of many supporting tissues during development and its origins in primates are not well understood. One previous study demonstrated differentiation of human ExM from naive pluripotent stem cells (Pham et al, cited by the authors); a recent work showed the dependence of this pathway on WNT signaling (<https://doi.org/10.1111/cpr.13761>) as did the authors' previous study (Ai et al), and ExM formation has been described in numerous post-implantation human embryo models though the origins of the tissue have not been deeply studied in this context. Novel aspects of this work include a rapid and efficient induction of ExM, the extensive comparison of ExM derived from naive and primed cells and the identification of a primitive streak like intermediate during ExM differentiation. Further differentiation of these cells or functional characterization (for example extracellular matrix formation) are not studied.

The findings regarding the presence of putative ExM precursor cells with a primitive streak phenotype are interesting, but the inferences regarding the primitive streak as an intermediate to ExM are potentially confusing and do not have a very strong experimental basis. The primitive streak is not a cellular state but a structure that emerges at the onset of gastrulation, and it is widely agreed that ExM is identifiable before this point in primates, namely around CS5. Indeed, this is a feature that distinguishes primate from mouse development. The authors study induction of ExM in 2D culture which as discussed below may not reflect a normal developmental sequence. If the authors are contending that a primitive streak structure appears prior to gastrulation to give rise to ExM, they need to provide more convincing microscopic images than those shown in Figure 4 and S4. A more likely interpretation of the authors' data is that the ExM intermediate recapitulates a gene expression program also used in formation of the primitive streak. However, it should also be noted that

scRNA-seq data do not prove lineage relationships; they are suggestive only. Clonal isolation in vitro of PS-like cells (using a cell surface antibody or reporter cell line) followed by a demonstration of their conversion to ExM could address this point more directly. See below for further considerations around this issue.

Response

We sincerely appreciate your insightful and constructive comments and suggestions.

The study primarily focuses on the developmental dynamics and signaling mechanisms underlying ExM specification from hESCs, as well as the influence of the initial pluripotent state on ExM specification. Further differentiation assays or functional characterizations of ExMs, which are beyond the scope of this research, will be investigated in future work.

We apologize for the confusion resulting from our inappropriate statements. We acknowledge that the claim that “the primitive streak as an intermediate to ExM” is not rigorous. In the revised manuscript, we incorporated your constructive suggestion that “ExM intermediate recapitulates a gene expression program also used in formation of the primitive streak” and redefined ExM precursor cells with a primitive streak-like phenotype as primitive streak-like intermediates (PSLIs). Additionally, we isolated MESP1⁺ PSLIs using flow cytometry and confirmed their ability to further differentiate into ExMs, providing direct evidence that PSLIs differentiate into ExMs (Figs. 6m–o, 7h–k in the revised manuscript).

Specific points

1. L154-has EnExM any counterpart in the normal embryo? Could its phenotype reflect hypoblast origin of ExM as has been hypothesized previously?

Response

Thanks, this is a very good question. EnExM has no counterpart in human embryo. Correlation analysis shows that EnExM shares similarities with both ExM and endoderm (Supplementary Fig. 8f in the revised manuscript), suggesting that its characteristics may lie intermediate between ExM and endoderm. Integration analysis of scRNA-seq datasets reveals that a small subset of EnExM cells resemble the VE/YE in cultured human embryos and the endoderm in in vivo gastrulating embryo, while the remaining EnExM cells are similar to the ExM1 in cultured human embryos

(Fig. 1j, k, Fig. 8e in the revised manuscript).

Fig. S1. Identification of endodermal cells. **a** Subclustering analysis of scRNA-seq data of EnExM cluster in Fig. 1f. UMAP plot is color-coded according to cell subcluster identity annotations. **b** Dot plots of candidate genes specific for the indicated cell subtypes. **c**, **d**, UMAP plots of the indicated genes expressed in EnExM (**c**) and differentiation time courses for CBN4 (**d**).

Within the EnExM cluster, we further identified three subclusters, with subcluster-2 exhibiting endodermal characteristics (data not shown in this manuscript; please see the above Fig. S1a–c). Consistently, we observed a small number of cells

expressing endoderm genes in CBNs across various time points (data not shown in this manuscript; please see the above Fig. S1d), but they could not be identified as a separate cluster (Fig. 3e in the revised manuscript). Bulk RNA-seq data also showed that there was no significant change in endodermal gene expression during the progression of CBNs (Fig. 3c in the revised manuscript), suggesting a minimal presence of endodermal cells within CBNs.

These results suggest that a small number of ExMs in CBNs may be generated via the hypoblast route, but we lack solid experimental evidence, and it is unclear whether these small amounts of endodermal cells are specified from PSLs, hence we did not draw conclusions in the manuscript. In future work, this question is well worth further exploration.

2. L168- in Reference 3 (Ai et al.), the authors clearly identified two populations of ExM-one that emerges early, at E11, before cells of the primitive streak which appear at E13-14, and one that appears later, contemporaneously with the PS (data from Fig. S1 Ai et al.). Is it possible that the ExM produced in the 2D cultures represents a later wave of ExM that actually originates from the primitive streak, and that the earlier population is not represented in the 2D model?

Response

Thanks, this is a very good question.

To compare the relationship between CBN cultures and various embryonic lineages, we integrated scRNA-seq data from cultured human embryos and CBNs across different time course and yielded congruent UMAP outputs (Fig. 4a in the revised manuscript). Correlation analysis and marker gene expression patterns exhibited prominent similarities in the following cluster pairs: ExM1 (embryo) versus ExM1 (CBN), and ExM2 (embryo) versus ExM2 (CBN) (Fig. 4a–c in the revised manuscript). In other words, the ExM1 and ExM2 in CBNs exhibit the highest similarity to the ExM1 and ExM2 in cultured human embryos, respectively. Thus, our 2D model generates counterparts of both ExM1 and ExM2 found in cultured human embryos.

3. L270 emergence of PS like cells on D1 of the protocol indicates that the 2D system is not reflective of any normal developmental sequence; naive epiblast does not directly convert to PS.

Response

Thanks, very insightful comments.

Indeed, naive epiblast does not directly convert to primitive streak. Therefore, following your constructive suggestion, in the revised manuscript, we refer to the ExM precursors with primitive streak-like gene expression patterns as PSLIs.

In the second paragraph of the "Discussion" section of the revised manuscript, we have added the following statement: Although naive hESCs exhibited a transient upregulation of primed genes such as *CD24*, *DPPA2*, and *ETV4* under CB conditions on D1, this was accompanied by the loss of pluripotency marker *SOX2* and an upregulation of PS genes, including *TBXT*, *MIXL1*, and *SP5* (Fig. 3e, g). A previous study has reported that the transition of naive hESCs to a stable primed state requires 10 days¹. Consequently, naive hESCs rapidly initiated differentiation in CB conditions, progressing from naive pluripotency to PSLIs and then to ExMs, prior to transitioning to the primed state (Figs. 3e, g, 5l). These results indicate that naive hESCs, despite not responding to direct germ layer induction¹, can differentiate into ExMs in response to exogenous signals via PSLIs, similar to their rapid differentiation into trophoblasts in response to MEK and TGF- β inhibitors^{2,3}.

Similarly, in cultured embryos, we also observed cells expressing primitive streak genes *T* and *SP5* as early as E10 in the EPI (Fig. 4e, f in the revised manuscript), significantly earlier than the accepted onset of gastrulation. Therefore, during embryonic development, the generation of PSLIs may not require the EPI to undergo a naive-to-primed pluripotency transition, contrasting with primitive streak formation.

4. L283 what is the relationship if any between ExM1 and ExM2 described in this study and the populations with the same names in the embryo study in Ai et al. (point 1 above).

Response

Thanks for the question. Please see 'Response 2'.

5. L311-this is a confusing statement, see FigS1 of Ai et al. RNA velocity does not directly establish lineage relationships.

Response

Thanks for the constructive comment. In the revised manuscript, we have

removed this statement.

6. L317-as noted above, the 2D system, though an efficient means of differentiating ExM, does not reflect a normal developmental sequence.

Response

Thanks for the constructive comment.

We have recognized that the use of "primitive streak" to denote "ExM intermediates" in the original manuscript was inappropriate. In the revised manuscript, we have corrected the designation of ExM intermediates from PS to PSLI. However, our scRNA-seq data indeed exhibited prominent similarities between cultured human embryos and CBNs in the following cluster pairs: PostE-EPI versus Tra, PS1 versus PSLI1, PS2 versus PSLI2, ExM1 (embryo) versus ExM1 (CBN), and ExM2 (embryo) versus ExM2 (CBN) (Fig. 4a–c in the revised manuscript).

Additionally, in cultured human embryos, it remains to be determined whether the previously defined PS populations, PS1 and PS2⁴, are true primitive streak cells or potential PSLIs. Therefore, in the third paragraph of the “Comparable developmental dynamics of ExMs between cultured human embryos and CBNs” section of the revised manuscript, we have added the following statement: We previously identified the presence of PS cells (PS1 and PS2) as early as E11 in cultured human embryos using scRNA-seq⁴. Consistently, a recent study also detected T⁺ PS cells in cultured human embryos at E12 through IF staining⁵. These so-called PS cells^{4,5} appear earlier than the acknowledged onset of gastrulation, thus their true identity—whether as PS cells or PSLIs (ExM precursors)—remains an open question.

7. L329-342-the conclusion that ExM arises from PS in the embryo at E11 (PS was present at very low levels in the authors’ previous study Ai et al. at this time point) is based on 2 embryos and a handful of T⁺ cells (scattered around epiblast) a subset of which co-express ExM markers. These data may be consistent with the presence of cells precociously expressing primitive streak markers at E11 but do not convincingly illustrate the presence of a primitive streak. It would be informative to see and E13 embryo similarly stained for comparison.

Response

Thanks for your very good comment and suggestion.

Sorry for the confusion caused by the inappropriate statement. Indeed, we did not observe the presence of the primitive streak structure in the E11 embryos. We only

found a small number of cells expressing T, which we have referred to as PSLIs in the revised manuscript (Fig. 4g and Supplementary Fig. 4d in the revised manuscript).

In the E14 embryo, we observed a PS-like structure and nearby ExMs (data not shown in this manuscript; please see the following Fig. S2), but without lineage tracing data, we cannot confirm whether these ExMs originate from PS. In our previous study, abundant ExMs were detected in E13 embryos through IF staining (Please refer to ‘Supplementary information, Fig. S1f, g’ in the reference⁴). Due to the scarcity of human embryos, we did not perform further staining analysis on E13 embryos in this study. Based on our experimental data, we can only conclude that the formation of intermediates with a PS-like phenotype precedes the emergence of ExMs.

Fig. S2. Staining images showing that ExMs appear in the proximal region of the primitive streak-like structure (PSLS) in E14 cultured human embryos. White numbers indicate section (S) numbers. Solid and dashed lines indicate PSLs and ExMs, respectively.

8. L338-again the statement that no ExM is present at E11, but that PS is, contradicts much prior data including that in Ai et al.

Response

Sorry for the confusion caused by the inaccurate presentation in the original manuscript. To address the seemingly contradictory statement, we provide three explanations:

(1) In the revised manuscript, we have redefined the ExM intermediate as PSLI. PSLI is not PS; it merely shares similar gene expression characteristics with PS cells.

(2) The PS (PS1 and PS2) cells we previously identified in cultured human embryos may actually be PSLIs, this was discussed in the third paragraph of the "Comparable developmental dynamics of ExMs between cultured human embryos and CBNs" section of the revised manuscript.

(3) Indeed, our previous scRNA-seq data analysis showed that PS and ExM first emerge in E11 embryos (Please refer to 'Supplementary information, Fig. S1c' in the reference⁴), but this result is based on cell cluster classification rather than gene expression. In fact, our scRNA-seq data show that PSLIs expressing *T* and *SP5* are detected as early as E10 in the EPI, while ExMs have not yet appeared at this stage (Fig. 4d–f in the revised manuscript). For details, please see the first paragraph of the "Comparable developmental dynamics of ExMs between cultured human embryos and CBNs" section of the revised manuscript.

9. L448-548-the differences between the trajectories of naive vs primed cells are clearly documented, but the developmental significance of these findings is unclear.

Response

Thanks, this is a very good question.

Our results indicate that the initial pluripotent state influences ExM specification from hESCs in several aspects, including signal response, cellular composition, developmental progression, and transcriptional characteristics of the resulting ExMs, which results in different developmental trajectories between naive and primed hESCs under CB or CBA conditions. These findings may imply that (1) ExM specification from human epiblast is spatiotemporally specific in signaling response and developmental progression, and (2) there are transcriptional and functional differences in the ExMs derived from human epiblast at different developmental stages. In the revised manuscript, we have discussed this topic. For details, please see the second paragraph of the "Discussion" section of the revised manuscript, where we have discussed this issue.

10. It is possible given the data presented here that a small transient population of cells with a primitive streak like phenotype give rise to ExM. This is feasible; it is certainly conceivable that the molecular program for embryonic mesoderm and ExM induction is similar. It is also possible that a subpopulation of ExM derives from PS after around E13. Perhaps the data are consistent with either interpretation, but they do not prove these possibilities directly. I do not think the authors can assert that all

ExM derives from the primitive streak, if that is what they are meaning to claim.

Response

Thanks for the insightful and constructive comments.

We cannot claim that all ExMs derive from the primitive streak. Following your suggestion, we have redefined the ExM intermediates as PSLIs in the revised manuscript. We propose that the specification of ExMs from hESCs predominantly involves a progression from pluripotency to PSLI to ExM.

Comments from reviewer(s): Reviewer #3 (Remarks to the Author):

In this study, Niu et al develop a new protocol to generate extra-embryonic mesoderm (ExM) cells from human ESCs and identify the signals that drive ExM fate from different pluripotent states. The biology of human ExM is poorly understood. Only very recent publications have begun shedding light on the mechanisms that control ExM specification. Therefore, this article will be of interest to the community. I support the publication of the article, but several issues need to be addressed.

Response

We would sincerely like to thank you for the encouraging comments on our manuscript.

Main comments:

The authors use AIC-N to maintain human ESCs in a naive pluripotent state, and AIC to preserve primed pluripotency. While these culture conditions have already been characterized by the authors, they are not broadly used by the community. Therefore, it would be important to first briefly explain them in the text (only the original references are provided), and to validate the key findings of the manuscript using alternative naive and primed conditions. This would strengthen the relevance of the findings.

Response

Thanks for your very good suggestion.

Following your suggestion, we have made the following improvements in the revised manuscript: (1) a brief explanation to AIC (for primed hESCs) and AIC-N (for naive hESCs) media. (2) validation of our findings using hESCs cultured in alternative naive (PXGL⁶) and primed (E8/mTeSR1) conditions.

Under CB and CBA conditions, naive hESCs cultured in PXGL and our AIC-N medium exhibited highly consistent phenotypes, both efficiently differentiating into ExMs (Fig. 7g, Supplementary Figs. 1k–m, 7h in the revised manuscript).

Under CB conditions, primed hESCs cultured in mTeSR1 and our AIC medium predominantly differentiated into AM-like cells. Although primed hESCs cultured in E8 also produced a large number of AM-like cells in CB conditions, they differentiated into significantly more (approximately half) ExMs compared to those cultured in AIC and mTeSR1 media (Fig. 5n and Supplementary Fig. 1k in the revised manuscript). Because the basal media of AIC and mTeSR are similar but differ significantly from that of E8⁷, we speculate that the composition of basal medium also affects the response of hESCs to exogenous signals, but this needs to be confirmed by

further studies

Together, we largely validated the study's key findings using alternative naive and primed hESCs in the revised manuscript, although there is a difference in the response to CB conditions between hESCs cultured in E8 and AIC/mTeSR1.

Figure 1: The authors use CB to induce ExM from human ESCs, and then human TSC medium for long-term culture. It is not clear whether CB can sustain the expansion of ExM. If yes, do the ExM cells retain their earlier ExM identity in this condition?

Response

Thanks, this is a very good question.

CB conditions can support the expansion of CBNs but appear unable to retain their earlier ExM identity. When cultured to passage 5 (P5) under CB conditions, these CBNs, termed CBN^{CB} cells, still maintain self-renewal and express common ExM markers, suggesting their ExM identity. However, compared to CBNs transferred to hTS-M, CBN^{CB} cells exhibit a distinct developmental trajectory and gene expression profile, with heterogeneity observed across different cell lines (data not shown in this manuscript; please see the following Supplementary Fig. 3).

Fig. S3. Extended culture of CBNs under CB conditions. **a** Schematic of extended culture of CBNs under CB conditions. **b** Brightfield and staining images for the indicated markers in extended cultured CBNs. Scale bars, 100 μm . CBN^{CB} , CBNs expanded under CB conditions; P, passage. **c** Principal-component analysis (PCA) of bulk RNA-seq data from differentiation time courses of CBNs under the indicated conditions, computed using the genes with FPKM ≥ 1 in at least one sample. **d** Spearman correlation analysis of gene expression patterns of the indicated samples. **e** Heatmap of representative pluripotency and ExM marker genes in naive hESCs and CBNs. Values represent $\log_2(\text{FPKM}+1)$ scaled by gene expression across samples.

In this study, we successively employed CB and CBA as induction systems for ExM specification, and hTS-M is a known medium for ExM expansion. To maintain consistency in expansion medium of ExMs induced from both CB and CBA conditions and to minimize confusion in the manuscript, we did not include these data in the revised manuscript (the above Supplementary Fig. 3). In future work, we will systematically compare the effects of different culture systems on ExM expansion and strive to identify media that can stably maintain early ExM identity.

Figure 2: The finding that Wnt signalling needs to be inhibited to make trophoblast is surprising, as Wnt activation is needed to preserve TSC identity. How is this explained? Are the trophoblast cells eventually differentiating to an extra-villous trophoblast identity when IWP2 is used (prolonged IBN culture)? Do the cells acquire a bona fide TSC identity if instead they are switched to human TSC medium after 4 days in IBN?

Response

Very insightful questions. Sorry for the confusion caused by the our inappropriate statements.

In the presence of BMP4, WNT inhibition leads to the progression of naive hESCs towards an AM-like rather than a trophoblast-like fate. This has been corrected in the revised manuscript.

Indeed, AM-like cells in hTS-M gradually shift towards a trophoblast fate^{4,8}. During the expansion of human trophoblast stem cells, WNT inhibition induces their differentiation into an extra-villous trophoblast identity^{9,10}. Since our focus is on ExM specification, the differentiation of AM-like or trophoblast-like cells is beyond the scope of this study. Consequently, we did not further investigate the plasticity of the AM-like cells obtained in this study.

Figure 4: the human embryo data is weak. The images provided are saturated. It is difficult to understand which structures are present, and the embryos seem to be disorganized. Some of the nuclei that are highlighted in Figure S4D seem to be DAPI fragments/debris, and all the arrowheads obstruct the view. Moreover, data from human embryos at day 13 showing ExM cells is needed to fully characterize ExM specification in in vitro cultured embryos. Finally, it is not clear how the authors put their findings in the context of embryo development. In the discussion, they mention “precocious gastrulating cells”. What does this mean? Either there is gastrulation or there isn’t. Do the authors see the hallmarks of gastrulation already on day 11? Is

there basement membrane breakdown? Is there delamination and loss of epithelial integrity of epiblast cells? Does this happen in a polarised manner within the epiblast? What is the meaning of the Brachyury+ cells that are detected on day 11?

Response

Thanks for the insightful and constructive comments.

In the revised manuscript, following your comments, we made the following changes: (1) reducing the fluorescence intensity of the staining images and downsized the arrowheads used for indication; (2) correcting the definition of PS to PSLI; and (3) removing the term “precocious gastrulating cells” from the ‘Discussion’ section.

In our previous study, abundant ExMs were detected in E13 embryos through staining (Please refer to ‘Supplementary information, Fig. S1f, g’ in the reference⁴). Due to the scarcity of human embryos, we did not perform further staining analysis on E13 embryos in this study.

The data presented here indicate that in cultured human embryos, precursor cells exhibiting PS-like gene expression patterns (termed PSLIs) are specified prior to ExM formation. This suggests that some ExMs in pre-gastrulation embryos may originate from EPI-derived PSLIs, which is consistent with our findings in CBNs.

See also “Responses 6–8 to Reviewer #2”.

Figure 5: in the CBP condition the authors mention that cells undergo an amnion to a trophoblast fate transition. Similarly to my previous comments, are bona fide trophoblast cells being specified? If switched to a human TSC medium, can they be maintained long-term?

Response

Thanks for the insightful questions. Sorry for the confusion arising from our inappropriate statement.

In the revised manuscript, we removed the statement “*suggesting that AM2 may represent a group of cells undergoing the transition from AM to TrB*” from the original manuscript.

Under CBP conditions, primed hESCs differentiate into two amnion-like cell types, AM1 and AM2. While these two cell types exhibit some differences, neither expresses the trophoblast (TrB) markers *HAVCR1*, *GCM1*, and *CGA* (Supplementary Fig. 5d in the revised manuscript), indicating that they have not acquired a trophoblast identity. Since this study focuses on ExM specification, and AM or TrB development

is not the focus of this research, we did not further investigate the cellular characteristics and developmental potential of AM1 and AM2.

Generally, throughout the text, amnion and trophoblast are grouped together as one fate, alternative to the ExM. Why are they grouped? Is this because both cell types are specified? Or is it because technically the authors have not been able to distinguish whether the cells are amnion or trophoblast, or even a mixed identity?

Response

Thanks for the insightful questions. Sorry for the confusion arising from our inappropriate statement.

In the revised manuscript, we removed the statement “TrB” from the original manuscript.

The finding that primed cells require Activin-A for ExM specification while naive cells don't is surprising, as it would be likely that naive cells transition through a primed state before making ExM (potentially post-EPI state in the UMAP plot (Figure 5J)). Is this the case? It would be informative to compare the dynamics of ExM induction from naive and primed to better understand the findings.

Response

Thanks for the insightful comments and questions.

In this study, we found that during ExM specification from hESCs, primed cells require Activin-A, but naive cells do not. Although the molecular mechanisms underlying the differential signaling responses caused by the initial pluripotent state remain unclear, the inability of naive cells to transition to the primed state prior to specification into PSLIs or ExMs may be a significant factor contributing to these differences. We have discussed this issue in the second paragraph of the "Discussion" section of the revised manuscript, as detailed below: Although naive hESCs exhibited a transient upregulation of primed genes such as *CD24*, *DPPA2*, and *ETV4* under CB conditions on D1, this was accompanied by the loss of pluripotency marker *SOX2* and an upregulation of PS genes, including *TBXT*, *MIXL1*, and *SP5* (Fig. 3e, g). A previous study has reported that the transition of naive hESCs to a stable primed state requires 10 days¹. Consequently, naive hESCs rapidly initiated differentiation in CB conditions, progressing from naive pluripotency to PSLIs and then to ExMs, prior to transitioning to the primed state (Figs. 3e, g, 5l). These results indicate that naive hESCs, despite not responding to direct germ layer induction¹, can differentiate into

ExMs in response to exogenous signals via PSLIs, similar to their rapid differentiation into trophoblasts in response to MEK and TGF- β inhibitors^{2,3}.

Following your suggestion, we integrated scRNA-seq datasets between CBNs and CBPs across different time courses (Fig. 5I in the revised manuscript), finding that naive cells do not transition to the primed state prior to converting into PSLIs and ExMs.

Does the addition of Activin-A block the effect of IWP2?

Response

Thanks for raising this interesting question. Yes, the addition of Activin-A blocks the effect of IWP2.

Under IB (IWP2 + BMP4) conditions, the addition of Activin-A blocks the acquisition of the AM fate in hESCs, which largely retain their pluripotency regardless of whether they are in the naive or primed state (Supplementary Figs. 6I, 7I in the revised manuscript).

Figure 7: the description of the findings should be more conceptual. The authors provide lists of genes and lists of GO terms that globally lack meaningful information. What is the conclusion that is drawn from all the GO terms and differentially expressed genes? This also applies to the description of other figures but is especially relevant for Figure 7.

Response

Thanks for the constructive comments.

In the revised manuscript, we provided additional data and revised our description based on your constructive suggestion. For details, please refer to the section “Naive and primed hESCs show both shared and distinct aspects in ExM specification”, Fig. 8 and Supplementary Fig. 8.

Methodology: several immunofluorescence images appear saturated, and it is difficult to see any clear pattern of localization (for example active beta-catenin in panel 2C). Moreover, the quantifications provided are not appropriate (i.e., panel 2B). Fluorescence intensity levels need to be measured in single nuclei using unsaturated images and the data needs to be statistically analysed. This would also serve as a validation of the experimental tools. For example, is there a positive correlation between the nuclear fluorescence intensity of beta-catenin and H2B-GFP (from the

TCF/LEF:H2B-GFP line)?

Response

Thanks for the constructive suggestions.

In the revised manuscript, following your constructive suggestion, we made the following changes: (1) reducing the fluorescence intensity of the immunofluorescence images (Fig. 2a, e); and (2) quantifying the average fluorescence intensity in individual cells (Fig. 2c, f).

For Fig. 2b, in addition to presenting fluorescence intensity, it is more important to show the subcellular localization of the indicated markers (DAPI or transcription factor localizes to the nucleus).

For details, please see “WNT signaling is critical for ExM specification from naive hESCs” section, the first paragraph.

Minor comments:

Line 143-144: this sentence is incorrect. ExM cells have never been derived from a human embryo.

Response

Thank you for your reminder. In the revised manuscript, we have corrected this error as follows:

In the original manuscript: These results suggest that the D4 CBNs have similar characteristics compared to previously reported embryo- or naive hESC-derived ExMs.

In the revised manuscript: These results suggest that D4 CBNs have similar characteristics compared to previously reported ExMs in 3D-cultured human embryos or those derived from naive hESCs.

Where do PGCs come from based on the pseudotime analysis?

Response

Thanks for your question. Pseudotime analysis indicates that both PSLI1 and PSLI2 appear to contribute to the origin of PGCs (Supplementary Fig. 3h in the revised manuscript).

Some panels are not well organized, are difficult to interpret, or are not properly explained (either in the text or the legends). For example, in Figure 2A it is not clear if

the magnifications are done in any specific area of the non-magnified images. The non-magnified images are not very informative. Figure S3c mentions hrGFP-labeled female ESCs, what are these cells? What are they used for? Similarly, why are the authors using mCherry-labelled male cells for the experiments? Some UMAPs are missing axes. There is no consistency in the nomenclature. For example, in Figure 5B the authors refer to “primed cells” but in Figure 5C they refer to the same cells as “AIC”.

Response

Thanks for the constructive suggestion and reminder.

In the revised manuscript, we have inspected all panels and provided as detailed explanations as possible in the text or legends. Additionally, we have ensured consistent cell nomenclature and added UMAP axes.

For Fig. 2a, we have marked the magnified areas with red boxes and described them in the figure legend. Considering that magnified images often fail to represent the overall situation, we present the overall experimental phenotype using non-magnified images.

In this study, we found that naive and primed hESC lines of different sexes exhibited good consistency and repeatability under CB or CBA conditions. Additionally, we collected scRNA-seq data from many samples. To reduce batch effects and costs, we collected the indicated sample pair (naive and primed hESCs treated with CB or CBA for the same duration) as pooled sample by mixing equal cell numbers. The choice to pool samples with different fluorescent labels and sexes was to facilitate subsequent scRNA-seq data deconvolution. In the revised manuscript, we have explained this in the figure legends and methods.

Why one more day was used for the protocols that entailed the addition of Activin-A?

Response

Thanks for your careful observation.

Under CBA conditions, primed hESCs treated for four days still retained a small number of pluripotent cells (Supplementary Fig. 6a in the revised manuscript). Therefore, we extended the differentiation time by one day, which has been described in the text. To maintain consistency with primed hESCs, naive hESCs were also induced for 5 days.

Reference

- 1 Rostovskaya, M., Stirparo, G. G. & Smith, A. Capacitation of human naive pluripotent stem cells for multi-lineage differentiation. *Development* **146**, doi:10.1242/dev.172916 (2019).
- 2 Guo, G. *et al.* Human naive epiblast cells possess unrestricted lineage potential. *Cell stem cell* **28**, 1040-1056 e1046, doi:10.1016/j.stem.2021.02.025 (2021).
- 3 Io, S. *et al.* Capturing human trophoblast development with naive pluripotent stem cells in vitro. *Cell stem cell* **28**, 1023-1039 e1013, doi:10.1016/j.stem.2021.03.013 (2021).
- 4 Ai, Z. *et al.* Dissecting peri-implantation development using cultured human embryos and embryo-like assembloids. *Cell Res* **33**, 661-678, doi:10.1038/s41422-023-00846-8 (2023).
- 5 De Santis, R. *et al.* The emergence of human gastrulation upon in vitro attachment. *Stem Cell Reports* **19**, 41-53, doi:10.1016/j.stemcr.2023.11.005 (2024).
- 6 Bredenkamp, N. *et al.* Wnt Inhibition Facilitates RNA-Mediated Reprogramming of Human Somatic Cells to Naive Pluripotency. *Stem Cell Reports* **13**, 1083-1098, doi:10.1016/j.stemcr.2019.10.009 (2019).
- 7 Ai, Z. *et al.* Modulation of Wnt and Activin/Nodal supports efficient derivation, cloning and suspension expansion of human pluripotent stem cells. *Biomaterials* **249**, 120015, doi:10.1016/j.biomaterials.2020.120015 (2020).
- 8 Soncin, F. *et al.* Derivation of functional trophoblast stem cells from primed human pluripotent stem cells. *Stem Cell Reports* **17**, 1303-1317, doi:10.1016/j.stemcr.2022.04.013 (2022).
- 9 Siriwardena, D. *et al.* Marmoset and human trophoblast stem cells differ in signaling requirements and recapitulate divergent modes of trophoblast invasion. *Cell stem cell* **31**, 1427-1446 e1428, doi:10.1016/j.stem.2024.09.004 (2024).
- 10 Shukla, V. *et al.* NOTUM-mediated WNT silencing drives extravillous trophoblast cell lineage development. *Proc Natl Acad Sci U S A* **121**, e2403003121, doi:10.1073/pnas.2403003121 (2024).

Dear Reviewers,

On behalf of my co-authors, we appreciate you very much for the constructive comments and suggestions on our manuscript entitled “**Deciphering signaling mechanisms and developmental dynamics in extraembryonic mesoderm specification from hESCs**”. Your comments and suggestions are highly insightful and enable us to greatly improve the quality of our manuscript.

In the revised manuscript, we provided additional data to address the main concerns raised by you, and made revisions to the previous manuscript based on your suggestions.

The “**Point-by-point response to the reviewers’ comments**” was presented in the following:

Rebuttal: Manuscript NCOMMS-24-63802A

We would like to sincerely thank the reviewers for their constructive comments and suggestions, which we have used as the basis for revising our manuscript.

Comments from reviewer(s): Reviewer #2 (Remarks to the Author):

In this revised manuscript, the authors have provided some additional experimental findings, in particular direct fate analysis MESP1⁺ cells, and new analyses of scRNA-seq data, and have refined some of the interpretations of their data so as improve the precision of the conclusions of the study. While questions remain over the the origin and function of the ExM populations, the authors' work will inform future studies in this rapidly evolving field.

Response

We sincerely appreciate your encouraging comments.

Comments from reviewer(s): Reviewer #3 (Remarks to the Author):

The authors have made a significant effort to address the reviewers' comments. Most of the comments are successfully addressed, but the relevance of the findings for embryo development is still questionable. As it stands, the human embryo data is very weak, and no conclusions regarding the physiological relevance of the findings can be drawn. The authors have not provided additional experiments, as requested both by reviewer 2 and me. Either the authors perform additional experiments in human embryos to fully characterize what is happening in the embryo in terms of primitive streak and ExM differentiation, or they remove the embryo data and their claims of physiological relevance.

Response

We sincerely appreciate your encouraging comments and constructive suggestions.

Regarding the physiological relevance of primitive streak (PS) and ExM differentiation, as you pointed out, our human embryo data are indeed weak. Although we demonstrated in our previous response a high abundance of ExMs in the proximal region of the PS-like structure (PSLS) in 3D-cultured E14 embryos (re-presented in Figure R1 below), we acknowledge that ethical and technical constraints preclude us from providing lineage-tracing evidence to confirm whether

ExM in 3D-cultured human embryos originates from the PSLs.

Figure R1. Staining images showing that ExMs appear in the proximal region of the primitive streak-like structure (PSLS) in E14 cultured human embryos. White numbers indicate section (S) numbers. Solid and dashed lines indicate PSLs and ExMs, respectively.

We would like to clarify the following points regarding this study:

(1) Definition of PS-like intermediates (PSLIs): PSLIs are defined based on gene expression rather than physiological structure. Therefore, they are not equivalent to the PS but represent a cell population exhibiting a gene expression profile similar to that of the PS.

(2) Identification of ExMs in human E13 embryos: We concur that your suggestion, along with that of Reviewer 2, is highly valuable. Our previous work has already demonstrated the widespread presence of ExMs in human E13 embryos (Figure R2 below)¹. Given the scarcity of human embryos, we opted not to perform additional experiments. We sincerely hope for your understanding.

(3) Temporal relationship between PSLIs and ExMs: Based on immunofluorescence staining and single-cell RNA sequencing data, we propose that in 3D-cultured human embryos, potential PSLIs expressing PS-related marker genes appear prior to ExMs, rather than suggesting that a PS with a typical physiological structure precedes the emergence of ExMs. Due to the absence of lineage-tracing evidence, we have not established whether the potential PSLIs in 3D-cultured human embryos further specify into ExMs.

In summary, our embryo data do not establish a physiological relevance between PS and ExM differentiation, nor do they confirm that PSLIs invariably specify into

ExMs. However, our results demonstrate that in cultured embryos, PSLIs with PS-like gene expression profiles are specified prior to ExM formation. This suggests the possibility that PSLIs may be one origin of early ExMs. Therefore, we believe the embryo data in this study are worth retaining.

In the revised manuscript, we have removed any mention of human embryo experiments from the abstract and substantially toned down the relevant statements related to human embryo data. For instance, we have changed the section title in the Results from “Comparable developmental dynamics of ExMs between cultured human embryos and CBNs” to “PSLIs are specified prior to ExM formation in cultured human embryos” and used terms such as “potential” or “possible” in relevant statements.

Figure R2 (Reprinted from Supplementary information, Fig. S1 in our previous work¹). **f** and **g** immunofluorescence staining of epiblast, extraembryonic endoderm and ExM markers in three E13 extended cultured human embryos. White ordinal numbers indicate section numbers, arrowheads indicate ExMs, and red arrowheads indicate few OCT4⁻GATA6⁻KDR⁺ ExMs. **f** a twin embryo; **g** two singleton embryos. We cultured ten E6 blastocysts and obtained three normally developed E13 embryos for immunostaining.

In addition, there are a couple of points that need further clarification.

1. Response of other naïve and primed human ESCs to CB medium: the authors have addressed my initial comment. However, Figure 5n is missing naïve ESCs (AIC and PXGL) as a control. This is needed to conclude that E8-cultured primed ESCs behave more similarly to mTESR/AIC primed human ESCs than to naïve ESCs.

2. Methods: the authors mention that they have “reduced the fluorescence intensity of the immunofluorescence images”. How did they achieve that? A saturated image cannot be converted into an unsaturated image. It seems they did not repeat the experiments, which would be needed if the raw images were saturated as

quantifications would not be possible. The methods section should also include details on how the quantification of the immunofluorescence images was done.

Response

We sincerely appreciate your patience, rigor, and dedication in reviewing our manuscript.

We apologize for our oversight. The data from naive hESCs (AIC-N and PXGL) have now been included in the updated Figure 5n in the revised manuscript.

Regarding the saturation of the immunofluorescence images, we apologize for the lack of clarity in our previous response. Our raw images are not saturated, and thus, quantification of fluorescence intensity is not compromised. The processed images that appeared saturated in the original manuscript were due to excessive brightness adjustments using LAS X software or the LUTs function in NIS-Elements Viewer software. This issue has been corrected in the revised manuscript.

We have included detailed methods for the quantification of immunofluorescence images in the current revised manuscript (please refer to the Immunofluorescence staining section in the Methods).

Reference

- 1 Ai, Z. *et al.* Dissecting peri-implantation development using cultured human embryos and embryo-like assembloids. *Cell Res* **33**, 661-678, doi:10.1038/s41422-023-00846-8 (2023).